

# Volatile organic compounds at a rural site in Beijing: Influence of temporary emission control and wintertime heating

Weiqiang Yang[1,3], Yanli Zhang[1,2*], Xinming Wang[1,2,3*], Sheng Li[1,3], Ming Zhu[1,3],

Qingqing Yu[1,3], Guanghui Li[1,3], Zhonghui Huang[1,3], Huina Zhang[1,3], Zhenfeng Wu[1,3],

Wei Song[1], Jihua Tan[3], Min Shao[4,5]

State Key Laboratory of Organic Geochemistry and Guangdong Key Laboratory of

Environmental Protection and Resources Utilization, Guangzhou Institute of

Geochemistry, Chinese Academy of Sciences, Guangzhou 510640, China

Center for Excellence in Regional Atmospheric Environment, Institute of Urban

Environment, Chinese Academy of Sciences, Xiamen 361021, China

University of Chinese Academy of Sciences, Beijing 100049, China
State Joint Key Laboratory of Environmental Simulation and Pollution Control,

College of Environmental Sciences and Engineering, Peking University, Beijing

100871, China

Institute for Environmental and Climate Research, Jinan University, Guangzhou

511443, China

*corresponding authors:

Dr. Yanli Zhang

State key Laboratory of Organic Geochemistry,

Guangzhou Institute of Geochemistry,

Chinese Academy of Sciences, Guangzhou 510640, China



Tel.: +86-20-85292718; fax: +86-20-85290706.
E-mail: zhang_yl86@gig.ac.cn
Dr. Xinming Wang
State key Laboratory of Organic Geochemistry,
Guangzhou Institute of Geochemistry,
Chinese Academy of Sciences, Guangzhou 510640, China
Tel.: +86-20-85290180; fax: +86-20-85290706.
E-mail: wangxm@gig.ac.cn



**Abstract**
Secondary organic aerosols (SOA) contribute substantially to $PM_{2.5}$ during
wintertime severe haze events in north China, yet ambient volatile organic compounds
(VOCs) as SOA precursors are comparatively much less characterized in winter
especially in rural areas. In this study, ambient air samples were collected in 2014
from 25 October to 31 December at a rural site inside the campus of University of
Chinese Academy of Sciences (UCAS) in northeast Beijing for the analysis of VOCs.
Since that temporary intervention measures were implemented during 3-12 November
to improve air quality for the Asian-Pacific Economic Cooperation (APEC) summit
held in 5-11 November in Beijing, and that wintertime central heating started since 15
November in Beijing after the APEC summit, it is a good opportunity to study the
influence of the temporary control measures and the wintertime heating on the
ambient VOCs. As a result of temporary intervention measures implemented during
3-12 November (period II), total mixing ratios of non-methane hydrocarbons averaged
11.25 ppb, about 50% lower than that of 23.41 ppb before the APEC (25 October-2
November; Period I) or 21.71 ppb after the APEC (13 November-31 December;
Period III). Their ozone and SOA formation potentials decreased by ~50% and ~70%,
respectively, with the larger drop in SOA formation potentials attributed to more
effective control of aromatic hydrocarbons mainly from solvent use. Back trajectory
analysis revealed that the average mixing ratios of VOCs in the southerly air masses
were 2.3, 2.3 and 2.9 times that in the northerly ones during period I, II and III,
respectively; and all VOC episodes occurred under the influence of southerly winds,
suggesting much stronger emissions in the south urbanized regions than in the



northern rural areas. Based on the positive matrix factorization (PMF) receptor model,
changed contributions from traffic emission and solvent use could explain 47.9% and
37.6% of the reduction in ambient VOCs, respectively, during the Period II relative to
the Period I, indicating that the temporary control measures on vehicle emission and
solvent use were effective in lowering ambient levels of VOCs. Coal/biomass burning,
gasoline exhaust, and industrial emission were among the vital sources, and they
altogether contributed 60.3%, 78.6% and 78.7% of VOCs during the period I, II and
III, respectively. Coal/biomass burning, mostly residential coal burning, became the
dominant source which accounted for 45.1% of the VOCs during the wintertime
heating period, with a remarkably lower average contribution percentage (38.2%) in
the southerly air masses than that of 48.8% in the northerly air masses.



## 1. Introduction

Volatile organic compounds (VOCs) are precursors of tropospheric ozone and

secondary organic aerosols (SOA) (Forstner et al., 1997; Odum et al., 1997; Atkinson,
2000; O'Dowd et al., 2002; Sato et al., 2010). As SOA are important components of
$PM_{2.5}$ (particulate matter with an aerodynamic diameter less than 2.5 μm) (Cabada et
al., 2004; Lonati et al., 2005; Huang et al., 2014), reducing emission of VOCs would
benefit improving air quality in megacities, such as China's capital city Beijing,
where air pollution has become an extensive concern with increasing surface ozone
levels during summertime and severe $PM_{2.5}$ pollution during wintertime (Streets et al.,
2007; Ji et al., 2012; Wang et al., 2014). Since higher levels of ozone mostly occur
during summer and ozone formation in urban areas is largely VOC-limited (Shao et
al., 2009; Tang et al., 2010), many field measurements of VOCs in Beijing were
conducted in summertime with a focus on their sources (Song et al., 2007; Lu et al.,
2007; Yuan et al., 2009; Wang et al., 2010a) and their mixing ratios as well,
particularly during ozone episodes (Liu et al., 2009; An et al., 2012; Zhang et al.,
2012a; Liu et al., 2013). However, comparatively the role of VOCs in the wintertime
with $PM_{2.5}$ pollution is much less understood.

During extremely severe and persistent haze events in China, organic matter

(OM) could contribute 30-70% of the total $PM_{2.5}$, and higher fractions of SOA in OA
were observed during polluted days in winter in Beijing (Guo et al., 2014; Huang et
al., 2014; Zhang et al., 2014a). Therefore, the control of VOCs, as SOA precursors, is
also of great importance in the control of air pollution by $PM_{2.5}$ in wintertime. A





previous study demonstrated that levels of aromatic hydrocarbons and carbonyls
increased significantly under haze days in urban Beijing from 2008 to 2010 (Zhang et
al., 2014b), yet few reports are available about wintertime precursor VOCs in Beijing.
In urban areas, vehicle exhausts are important sources of SOA precursors (McDonald
et al., 2015; Liu et al., 2015a; Ortega et al., 2016; Deng et al., 2017; Gentner et al.,
2017). However, biomass/biofuel burning and coal burning may also contribute
substantially to SOA precursors (Yokelson et al., 2008; Shrivastava et al., 2015; Fang
et al., 2017), particularly in north China in wintertime when raw coal and biofuels are
widely used for household heating (Liu et al., 2016; Zhang et al., 2016a; Liu et al.,
2017). In fact, a study by Wang et al. (2013) in 2011-2012 revealed that even at an
urban site in Beijing coal combustion could account for 28-39% of VOCs observed in
ambient air. As raw coal and/or biofuel burning is widely occurring in the rural areas
in wintertime (Liu et al., 2016), it is necessary to investigate how the enhanced
emission due to wintertime household heating would influence the levels and
compositions of VOCs in rural areas, as forming SOA or ozone is an issue of regional
scale.
Due to a wide variety of emission sources of VOCs and large uncertainties of the
emission inventories of VOCs, to assess the effect of emission control measures on
reducing ambient VOCs is a highly challengeable task. The Chinese government has
implemented long-term pollution control actions and air quality has been greatly
improved in north China in recent years (Hao and Wang, 2005; Wang et al., 2009;
Zhang et al., 2012b; Liu et al., 2015b; Kelly and Zhu, 2016). However, air quality in



Beijing is not so satisfactory when compared to that in cities in the United States and
Europe, especially in wintertime with frequent haze events and high $PM_{2.5}$ levels.
Consequently, during critical international events such as the 2008 Olympic Games
(Wang et al., 2010b; Huang et al., 2010) and the 2014 Asia-Pacific Economic
Cooperation (APEC) summit, temporary intervention measures were adopted to
guarantee better air quality. This kind of temporary intervention provided a good
opportunity to study the effectiveness of various control measures on the reduction of
ambient air pollutants including VOCs (Yao et al., 2013; Huang et al., 2017). As for
the 21$^{th}$ Asia-Pacific Economic Cooperation (APEC) summit held in Beijing on 5-11
November 2014, temporary control measures in Beijing and its surrounding regions
resulted in significant drops of air pollutants including $PM_{2.5}$ and NOx (Huang et al.,
2015; Liu et al., 2015c; Wang et al., 2015; Xu et al., 2015; Zhang et al., 2016b); For
the VOCs in ambient air, as observed by Li et al. (2015) at an urban site inside the
campus of Peking University, total mixing ratios of VOCs reduced by 44% during the
APEC summit control period when compared to the period before. Since most
observation-based evaluations about the effectiveness of temporary emission control
measures were made with monitoring data in the urban areas, it is entirely necessary
to further investigate the influence in rural areas or a regional scale.

In this study, ambient air samples were collected at a rural site in the north of

Beijing from 25 October to 31 December 2014, covering the period with the enhanced
temporary emission control (3-12 November) for the APEC summit and the
wintertime heating period starting from 15 November. The objectives of present study



are: (1) to study changes in the mixing ratios and compositions of VOCs at a rural site
in Beijing in response to the emission control during the APEC summit and the
wintertime heating; (2) to identify crucial sources of VOCs in Beijing and their
changes during the PM-polluted wintertime; (3) to evaluate the impact of control
measures implemented during APEC summit on the reduction of VOCs in ambient air
in rural areas.
**2. Methodology**
*2.1 Sampling Site and Field Sampling*

The ambient air samples were collected at a site (40.41° N, 116.68° E; Fig. 1)

inside the campus of University of Chinese Academy of Science (UCAS) in Huairou
district of Beijing. The UCAS is located about 60 km northeast of central Beijing and
about 150 km northwest of the Tianjin city. It is surrounded by several small villages
and farmlands. The samples were collected 16 meters above ground on the top of a
four-story building, about 100 m west of a national road and only 1.5 km far away
from the APEC main conference hall.

Ambient air samples were collected from 25 October-31 December 2014 using

cleaned and evacuated 2 L silica-lined stainless steel canisters. During field sampling,
a model 910 canister sampler (Xonteck Inc., California, USA) with a constant flow
rate of 66.7 ml min$^{-1}$ was adopted to allow each canister to be filled in 60 min.
Samples were collected at approximately 10:00 and 15:00 of local time (LT) on sunny
days, one or two more samples were collected at 12:00 and/or 18:00 LT on haze days
when the visibility less than 10 km at relative humidity less than 90% (Fu et al., 2016).



A total of 153 samples were collected during sampling. According to the air pollution
control measures, the field campaign was divided into periods I (25 October-2
November), II (3-12 November) and III (13 November-31 December). Period II was
the time span when temporary control measures (Table 1;
http://www.zhb.gov.cn/gkml/hbb/qt /201411/t20141115_291482.htm) implemented
for better air quality. Wintertime heating started on 15 November just after the cease
of temporary control measures on 13 November. During the sampling periods,
prevailing winds were mostly from north to northwest (315-360°), the average wind
speeds were 3.5, 3.9, and 4.1 m s$^{-1}$, and average temperature was 11.4, 7.0, and 0.6°C
during periods I, II and III, respectively.
*2.2 Laboratory Analysis of VOCs and Carbon Monoxide*
All ambient air samples were analyzed with a Model 7100 pre-concentrator
(Entech Instruments Inc., California, USA) coupled with an Agilent 5973N gas
chromatography-mass selective detector/flame ionization detector (GC-MSD/FID,
Agilent Technologies, USA). Detailed cryogenically concentration steps are described
elsewhere (Zhang et al., 2012c). Briefly, 500 ml ambient air samples in the canister
were first pumped into the primarily trap with glass beads and then concentrated with
liquid-nitrogen cryogenic trap at -180°C. Following the primary trap was heated to
10°C, and all target compounds were transferred by pure helium to a secondary trap at
-50°C with Tenax-TA as adsorbents. Majority of H$_2$O and CO$_2$ were removed through
these two traps. The secondary trap then was heated to get VOCs transferred by
helium to a third cryo-focus trap at -170°C. After the focusing step, the third trap was



rapidly heated and the VOCs were transferred to the GC-MSD/FID system. The
mixture were first separated by a DB-1 capillary column (60 m×0.32 mm×1.0 μm,
Agilent Technologies, USA) with helium as carrier gas, and then split into two ways,
one is a PLOT-Q column (30 m×0.32 mm×20.0 μm, Agilent Technologies, USA)
followed by FID detector, another is to a 0.35 m×0.10 mm I.D. stainless steel line
followed by MSD detection. The GC oven temperature was programmed to be
initially at 10°C, holding for 3 min; next increased to 120°C at 5°C min$^{-1}$, and then
10°C min$^{-1}$ to 250°C with a final holding time of 7 min. The MSD was selected ion
monitoring (SIM) mode and the ionization method was electron impacting. Carbon
monoxide (CO) in the ambient air samples were also analyzed with an Agilent model
6890 gas chromatography equipped with a FID and a packed column (5Å Molecular
Sieve 60/80 mesh, 3 m×1/8 inch). CO was first separated by packed column, then
converted to $CH_4$ by Ni-based catalyst and finally detected by FID (Zhang et al.,
2016a).
*2.3 Quality Control and Quality Assurance*
Before sampling, all canisters were flushed at least five times by repeatedly
filling and evacuating humidified zero air. In order to check if there was any
contamination in the canisters, all canisters were evacuated after the cleaning
procedures, re-filled with pure nitrogen, stored in the laboratory for at least 24 h, and
then analyzed the same way as field samples to make sure that all the target VOC
compounds were not present.
Target compounds were identified based on their retention times and mass





spectra, and quantified by external calibration methods. The calibration standards
were prepared by dynamically diluting the Photochemical Assessment Monitoring
Stations (PAMS) standard mixture and TO-14 standard mixture (100 ppbv, Spectra
Gases Inc., New Jersey, USA) to 0.5, 1, 5, 15 and 30 ppb. The calibration curves were
obtained by running the five diluted standards plus humidified zero air the same way
as the field samples. The humidified zero air was initially analyzed every day to
ensure the cleanness of system and then the analytical system was challenged daily
with a one-point (typically 1 ppb) calibration before running air samples. If the
response was beyond +/-10% of the initial calibration curve, recalibration was
performed. The method detection limits (MDL) for each VOCs species were
presented in Table 2.
*2.4 Positive Matrix Factorization (PMF)*
PMF is a multivariate factor analysis tool that decomposes a matrix of sample
data into two matrices: factor contributions (G) and factor profiles (F). The method is
reviewed briefly here and described in greater detail elsewhere (Paatero and Tapper,
1994; Paatero, 1997). PMF uses both concentration and user-provided uncertainty
associated with the data to weight individual points. Data values below the MDL were
substituted with MDL/2; missing data values were substituted with median
concentrations. If the concentration is less than or equal to the MDL provided, the
uncertainty is calculated using the equation of $Unc = 5/6 \times MDL$; if the concentration
is greater than the MDL provided, the uncertainty is calculated as $Unc = [(\text{Error}$
$\text{faction} \times \text{mixing ratio})^2 + (MDL)^2]^{1/2}$. The number of factors in PMF was initially





chosen based on the result of PCA/APCS model (Zhang et al., 2012c).

## 3. Results and discussion

*3.1 Changing mixing ratios and compositions*
As mentioned above, during the period II (3-12 November) temporary emission
control measures were implemented to improve air quality during the 2014 APEC
summit. Total mixing ratios of VOCs observed at the rural site inside UCAS during
the period II was $11.25 \pm 3.22$ ppb in average, significantly lower than that of $23.41 \pm$
5.76 ppb during period I and $21.71 \pm 2.97$ ppb during period III (Fig. 2). These levels
were less than halves of 57.45, 36.17, and 56.56 ppb observed by Li et al. (2015) at an
urban site in Beijing before, during and after the APEC summit, respectively.
However, both our measurements at a rural site in this study and the measurements at
an urban site by Li et al. (2015) consistently demonstrated that the temporary
emission control resulted in a large decrease in ambient VOCs during the APEC
summit, with more than 30% reduction in the urban areas (Li et al., 2015) and about
50% reduction in rural areas as observed in this study. This reduced ambient mixing
ratios of VOCs during the period II was also in line with the decreased $PM_{2.5}$
concentrations observed in Beijing during the APEC summit (Liu et al., 2015c), or
reduced $NO_2$ vertical column densities (VCD) and aerosol optical depth (AOD) in
Beijing during the APEC summit based on remote sensing (Huang et al., 2015).
The percentages shared by alkanes, alkenes, and ethyne in total VOCs were quite
similar: alkanes accounted for 54, 57 and 54% of VOCs, alkenes accounted for 12, 16
and 17%, and ethyne accounted for 13, 14 and 14% of VOCs during periods I, II and





III, respectively. Instead, percentages shared by aromatics became lower during
period II (12%) when compared to that in period I (21%) or period III (15%).

The mean mixing ratios of alkanes, alkenes, aromatics and ethyne during period

II were 6.47, 1.83, 1.33, and 1.62 ppb (Fig. 2), and they decreased by 49.0, 32.5, 72.8,
and 48.1%, respectively, when compared to those during period I. Aromatics
evidently had a more substantial drop. Benzene, toluene, ethylbenzene, and
m,p-xylene, which are the most abundant aromatics and usually collectively termed as
BTEX, were 52.8, 73.1, 78.8, and 80.5% lower during period II than during period I,
respectively.

The total ozone formation potentials (OFPs) based on the maximum incremental

reactivity (Carter, 2009) in average during periods I, II and III were 60.64, 28.51, and
61.47 ppb (Table S1), respectively, with a 53.0% reduction during period II relative to
the period I (Fig. 2). Their secondary organic aerosol formation potentials (SOAFPs)
under high-NOx and low-NOx conditions (Ng et al., 2007; Lim and Ziemann, 2009)
were also calculated (Table S2). As showed in Fig. 2, total SOAFPs under low-NOx
conditions decreased by 71.0% from 8.77 $\mu g\ m^{-3}$ during the period I to 2.54 $\mu g\ m^{-3}$
during period II, and total SOAFPs under high-NOx conditions decreased by 64.4%
from 4.02 $\mu g\ m^{-3}$ during period I to 1.43 $\mu g\ m^{-3}$ during period II. This significant
decrease in OFPs and SOAFPs during period II is related to lowered VOCs mixing
ratios, especially larger drop in reactive alkenes and aromatics: alkenes and aromatics
explain 26% and 52% of the reduction in total OFPs, respectively; while the decrease
in total SOAFPs is mostly due to changed contribution by aromatic (Table S2), whose





SOAFPs decreased from 7.30 µg m$^{-3}$ during period I to 1.93 µg m$^{-3}$ during period II
under low-NOx condition, 2.39 µg m$^{-3}$ during period I to 0.75 µg m$^{-3}$ during period II
under high-NOx condition. Decrease the emissions of reactive alkenes and aromatics
are especially effective for OFPs and SOAFPs reduction.
*3.2 Pollution episodes and influence of source regions*

As showed in Fig. 3d and 3e, a number of episodes with mixing ratios of VOCs

over 30 ppb were recorded along with the increase in CO and SO$_2$ concentrations (Fig.
3d) during the campaign, such like that on 4-5 November, 15-16 November, 18-21
November, 28-30 November, 17 December, and 26-28 December. During the episode
on 3-5 November, for example, the total mixing ratio of VOCs was 14.30 ppb on 3
November, reached 31.96 ppb on 4 November, and then decreased again to 13.83 ppb
on 5 November. As shown in Fig. 3a, wind speeds were all below 2 m s$^{-1}$ during 3-5
November, and the planetary boundary layer (PBL) height on 4 November (477 m)
was approximately 83% of that on 3 November (578 m) (Fig. 3c). This lower PBL
height on 4 November could only partly explain the higher levels of VOCs. Figure
S1a showed the 72-h back trajectories (HYSPLIT, ver. 4.0; http://www.arl.noaa.
gov/ready/hysplit4.html) of air masses from 3-5 November at the height of 100 m in
12-h intervals and the corresponding mixing ratios of VOCs. It demonstrated that
mixing ratios of VOCs increased rapidly while air masses changed from the northerly
to the southerly, and then declined sharply while the air masses turned back from the
southerly to the northerly again. The southern areas of UCAS are the central Beijing
with stronger emissions, consequently air masses passed through these areas would

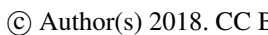



carry higher levels of pollutants to the sampling site, leading to the quick increase of
mixing ratios of VOCs. This rapid change of source regions could reasonably explain
more than the PBL height during the pollution episode of VOCs. As showed in Fig.
S1b, 1c, and 1d, back trajectories also suggested that the episodes on 18-21 November,
28-30 November and 26-28 December are related to the changed source regions.

According to the 72-h back trajectories, air masses arriving at the sampling site

could be categorized into two types (Fig. 4): 1) southerly (S) air masses, which passed
through Hebei, Shandong, Tianjin, and central Beijing with high-density emissions
before reaching UCAS; 2) northerly (N) air masses, which originated from Mongolia,
quickly passed through areas with less anthropogenic activity and low-density
emission before reaching UCAS. The pollution episodes with higher mixing ratios of
VOCs and CO, including the cases on 26-30 October, 4-5 November, 15-16
November, 18-20 November, 25-26 November and 26-28 December (Fig. 3d and 3e),
all occurred under the influence of southerly air masses, also suggesting the impacts
of emissions in the south.

During period I, II and III, the average mixing ratios of VOCs for southerly air

masses were 2.3, 2.3 and 2.9 times that for northerly air masses (Fig. 4), respectively;
OFPs in the southerly air masses were 2.0, 2.0 and 3.3 times that in the northerly air
masses, respectively; and SOAFPs in the air masses from the south were 1.7, 3.3, and
3.7 times that in the air masses from the north under low-NOx conditions, and 1.9, 2.7,
and 3.5 times that in the air masses from the north under high-NOx conditions,
respectively. This indicates that the north and south regions are completely different in





their source strengths. Developing strict control measures in the southern region is a
cost-effective way for abating VOCs pollution in Beijing. For the other cities suffered
from ozone and $PM_{2.5}$ pollution in the world, adopt a series of control measures in
VOCs hotspot are essential for pollution prevention.

As mentioned above, the mixing ratios of VOCs, as well as their OFPs and

SOAPFs, decreased greatly during period II. We can further see the changes in the
southerly and northerly air masses to indicate the changes in different source regions.
In the southerly air masses, when compared to that during period I average mixing
ratios of alkanes, alkenes, aromatics, and ethyne during period II were 8.32, 2.16, 1.93,
and 2.23 ppb, with reduction rates of 46.0, 33.3, 64.3, and 44.7%, respectively;
accordingly, OFPs decreased by 48.1% and SOAFPs decreased by 63.5 % (low-NOx
conditions) and 57.6% (high-NOx conditions) during period II when compared to that
during period I (Fig. 4). In the northerly air masses, average mixing ratios of alkanes,
alkenes, aromatics, and ethyne decreased 37.7, 4.8, 87.0, and 18.4% during period II
when compared to that during period I, respectively; OFPs decreased by 48.9% and
SOAFPs decreased by over 70% during period II relative to period I (Fig. 4). As
discussed below, more drastic decrease in aromatics in both the northerly and
southerly air masses implied more effective emission control in industrial solvent use
during the APEC summit, and the less changes in mixing ratios of alkenes in the
northerly air masses were related to the less effective control of domestic
coal/biomass burning in the northern regions. The mixing ratios of VOCs in the
southerly and northerly air masses during period III were 36.1% and 7.2% higher than





that during period I, respectively. This difference in the increase rates might be
explained by the fact that the urban areas in the south are largely central heating areas
where heating supply was only available since 15 November, and the northern areas
were largely rural areas where individual household heating might already started
during period I.
*3.3 Source attribution and apportioning*
*3.3.1 Indication from tracers*
The great changes in mixing ratios of VOCs during campaign might be resulted
from changed contribution by emission sources, such like enhanced emission control
during the APEC summit or intensified emission due to wintertime heating. These
changes could be indicated by characteristic fingerprints of different sources (Guo et
al., 2007).
The toluene/benzene (T/B) ratio, a widely used indicator for sources of aromatics,
was 1.09, 0.67 and 0.70 in average during period I, II and III, respectively (Fig. S2a).
While T/B ratios during periods II and III were approaching 0.6, which is
characteristic of coal/biomass burning (Liu et al., 2008; Liu et al., 2015d), the ratios
during the period I fell between that of coal/biomass burning (0.6) and vehicle exhaust
(1.6), which is characteristic of vehicular exhaust (Wang et al., 2002; Liu et al., 2009;
Zhang et al., 2013a). Carbon monoxide (CO), a typical tracer of incomplete
combustion of biomass or fossil fuels (Parrish et al., 2009; Zhang et al., 2015a),
showed highly significant correlations with benzene during the period II ($r^2$=0.96, Fig.
S2b) and the period III ($r^2$=0.88, Fig. S2b). SO$_2$, a good indicator of coal burning (Li





et al., 2017), had similar concentrations during periods II and periods I, but its
concentrations increased 56.5% in average during period III compared to that during
period I (Fig. 3d), suggesting that coal burning contributed more after the start of
central heating. Methyl tert-butyl ether (MTBE), a specific indicator of gasoline
related traffic emission (Song et al., 2007; Cai et al., 2010), showed better correlation
with benzene during period I ($r^2$=0.88, Fig. S2c) than during period II and III.
As toluene, ethylbenzene and xylene (TEX), are mainly from solvent use in
painting, decoration and coating (Guo et al., 2007; Zhang et al., 2012c), the ratios of
TEX to CO were widely used to examine the impact of solvent use relative to
combustion emissions (Zhang et al., 2013a). The ratios of T/CO, E/CO and X/CO
were 0.61±0.09, 0.23±0.06 and 0.35±0.07 (ppb/ppm) during period II, obviously
lower when compared to that of 1.16±0.49, 0.59±0.24 and 0.99±0.41 during period
I, or 1.34±0.27, 0.40±0.06 and 0.83±0.09 during period III (Fig. 5B), respectively.
This drop in aromatics/CO ratios during period II also reflected more effective control
of solvent use during the APEC summit.
If further categorized according to the air masses trajectories, the ratios of T/CO,
E/CO and X/CO decreased 29.5, 45.7 and 45.7% in the southerly air masses during
period II relative to period I, and decreased 68.0, 80.3 and 83.0% in the northerly air
masses during period II relative to period I, respectively (Fig. 5A). Apparently larger
decrease in TEX/CO ratios in the northerly air masses reflected the control of solvent
use was more effective in northern regions.
*3.3.2 Source Apportioning by PMF*





Thirty-five most abundant VOCs, including alkanes, alkenes, aromatics, ethyne,
and sources tracers such as chloromethane, trichloroethylene, tetrachloroethylene and
MTBE, plus $SO_2$ and CO, were selected for the PMF receptor model. Figure 6 shows
the 5 sources retrieved by the model.
Factor 1 has high values of MTBE and $C_5$-$C_6$ alkanes. MTBE is a common
gasoline additive in China and 2,2-dimethylbutane is used to enhance the octane
levels of gasoline (Chang et al., 2004; Song et al., 2007; Cai et al., 2010); Ethyne can
be formed during fuel combustion (Blake and Rowland, 1995; Song et al., 2007;
Suthawaree et al., 2010); $C_5$-$C_6$ alkanes are associated with unburned vehicular
emissions (Guo et al., 2004; Cai et al., 2010; Zhang et al, 2013b). Consequently factor
1 is related to the gasoline vehicle emission.
Factor 2 is distinguished by a strong presence of trichloroethylene,
tetrachloroethylene and moderate contributions of propene and butenes.
Trichloroethylene and tetrachloroethylene are species from manufacturing industrials
(Yuan et al., 2013; Zhang et al., 2015b); propene and butenes are gases widely used by
industries for make organic chemicals (Guo et al., 2007), such as production of
synthetic rubber in the petrochemical industry (Lau et al., 2010). Thus factor 2 was
identified as industrial emission.
Factor 3 accounts for a larger percentage of the toluene, ethylbenzene,
m/p-xylene and o-xylene. It is known that TEX are the primary constituents of solvent
(Guo et al., 2004; Yuan et al., 2009; Zheng et al., 2013; Zhang et al., 2014c; Ou et al.,
2015). They are also main component in emissions from auto factory painting and



building coating (Liu et al., 2008; Yuan et al., 2010). Therefore, this source is
considered as solvent use related to painting and architecture.
Factor 4 is diesel exhaust which is characterized by a significant amount of
n-undecane and n-dodecane (Song et al., 2007; Zhang et al., 2012c).
Factor 5 is characterized by the presence of ethane, ethylene, CO, $SO_2$ and
chloromethane. Chloromethane is the typical tracer of biomass burning (Liu et al.,
2008; Cai et al., 2010; Zhang et al., 2014c). Ethylene, ethane and propene are top 3
species of rice straw burning (Zhang et al, 2013c; Fang et al., 2017). The VOC species
from coal burning were mainly ethyne, $C_2$-$C_3$ alkenes and alkanes, and aromatics like
benzene (Liu et al., 2008). $SO_2$ is mainly from coal burning (Li et al., 2017). So factor
5 is related to the coal/biomass burning.
Figure 7 shows the source contributions during period I, II and III. During period
I gasoline exhaust was the largest source and accounted for 24.0% of VOCs, while
during period II coal/biomass burning became the largest source. The most significant
changes due to temporary emission control during the period II were in the
contribution percentages by coal/biomass burning (22.3% in period I and 42.4% in
period II) and by solvent use (21.9% in period I and 5.8% in period II). The large drop
in the contribution by solvent use was consistent with the above discussion about the
TEX/CO ratios. Quite similar contributions were observed for industrial emission and
diesel exhaust.
In the period III (13 November-31 December) with the central heating starting
from 15 November, coal/biomass burning became the largest source (45.1%), and





industrial emission, solvent use, diesel exhaust and gasoline exhaust accounted for
25.2, 12.8, 8.7 and 8.2% of VOCs, respectively. The time series of source
contributions during the campaign were showed in Fig. S3, the contribution
percentages by coal/biomass burning increased gradually with the wintertime heating,
while that of gasoline exhaust instead decreased.
Coal/biomass burning was an important source of VOCs during winter in Beijing,
especially during period III with the start of central heating. In Beijing, coal
consumption was greater than that of residential biomass (Liu et al., 2016). During
2008-2014 in Beijing the annual residential coal consumptions increased gradually
while the total coal consumption decreased (Beijing Municipal Bureau of Statistics,
2015). The residential coal combustion is prevailing for heating and cooking by using
domestic coal stoves in rural areas around urban Beijing particularly during
wintertime. In 2014, although the annual residential coal consumption accounts for 17%
$(2.93\times10^9$ kg a$^{-1}$) of the total coal consumption in Beijing (Beijing Municipal Bureau
of Statistics, 2015), residential coal burning could contribute predominately to
ambient VOCs from coal burning since the emission factors of VOCs from residential
coal burning have been found to be a factor of 20 greater than those from coal-fired
power plants (Liu et al., 2017).
Compared with that in the period I (Fig. S4), the contribution by solvent use
during the period II was reduced to a greater extent; it became 4.29 ppb lower and
could explain 37.6% of the reduction in ambient VOCs (Table S3). The contribution
by gasoline vehicles was 3.18 ppb lower and accounted for 27.9% of total reductions.



441 The contribution by diesel exhaust and industrial emission reduced 2.28 ppb and 1.35

442 ppb, and explained 20.0 and 11.8% of total reduction, respectively. Coal/biomass

443 burning showed similar contributions during period I and II with an elevated

444 contribution percentage in the period II due to the reduction in other sources. This is

445 consistent with the fact that during the APEC summit residential coal/biomass burning

446 was not restricted in the rural areas. Traffic-related sources (gasoline and diesel

447 vehicles) and solvent use account for 47.9 and 37.6% of total reduction in ambient

448 VOCs, indicating that control measures (Table 1) related to the control of traffic and

449 solvent use were among the most effective ways to reduce the ambient VOCs.

450  Figure 8 shows the source contributions in the southerly and northerly air masses

451 during period I, II and III, respectively. In the southerly air masses, traffic related

452 emission (gasoline and diesel vehicles) was the largest source, contributing 44.1 and

453 41.5% of VOCs during the period I and II, respectively; while coal/biomass burning

454 instead was the largest source during period III, contributing 38.2% of VOCs. In the

455 northerly air masses, coal/biomass burning contributed 28.8, 51.6 and 48.8% of VOCs

456 during period I, II and III, respectively. Overall, gasoline vehicle exhaust contributed

457 more VOCs in the southern regions (mostly densely populated urban areas) and

458 coal/biomass burning and diesel exhaust accounted for more emissions of VOCs in

459 northern regions (mostly rural areas). Contributions of different sources to most

460 reactive alkenes and aromatics based on PMF were presented in Fig. 9. Alkenes was

461 mainly coming from coal/biomass burning with shares of 31.2-68.0%, and gasoline

462 exhaust ranked second with contributions of 3.0-26.5%. Unlike alkenes, solvent use





was the major contributors of aromatics, accounting 77.5% during period I and 29.0%
during period II in the northerly air masses; gasoline exhaust contributed 8.2-43.6% of
aromatics during campaign. In the southerly air masses, reductions in solvent use,
gasoline exhaust, and diesel exhaust during the period II relative to the period I could
explain 38.1, 31.1, and 15.8% of total reduction of VOCs, respectively. In the
northerly air masses, reductions of solvent use, diesel exhaust, and gasoline exhaust
during the period II relative to the period I could explain 46.5, 35.8 and 11.9% of total
reduction of VOCs, respectively. Consequently, control measures related to solvent
use and gasoline exhaust were more effective in the southern regions, while the
control of solvent use and diesel exhaust emission were more effective in the northern
region.
*3.3.3 Source contributions to SOAFPs*

With the PMF source apportioning results, the contributions of SOAFPs by

different sources were further estimated. As showed in Fig. 10, under low-NOx
condition the SOAFPs by solvent use were much higher than that by other sources,
which were 4.88, 0.68 and 2.89 μg m$^{-3}$, accounting for 56.9, 27.2 and 54.7% of total
SOAFPs during period I, II and III, respectively. Gasoline exhaust contributed 19.2,
29.5 and 10.9%, and diesel exhaust contributed 16.5, 26.8 and 11.3% of SOAFPs
during period I, II and III, respectively. During the period II with temporary
intervention measures, the reduction of SOAFPs was mainly due to reduced
contribution by solvent use, gasoline exhaust and diesel exhaust, which could explain
69.1, 14.9 and 12.2% the reduction in SOAFPs, respectively. Under high-NOx



condition, calculated reduction of SOAFPs during the period II relative to the period I
could largely explained by reduced contributions by solvent use, diesel exhaust and
gasoline exhaust, which accounted for 54.0, 25.8 and 16.8% of the reduction in
SOAFPs, respectively.
It is worth noting that recent chamber studies revealed that aromatic
hydrocarbons or traditional VOCs could not fully explain SOA formed from
atmospheric aging of source emissions (Zhao et al., 2014; 2015; Liu et al., 2015a;
Deng et al., 2017; Fang et al., 2017), particularly for emissions from diesel vehicles or
biomass burning (Zhao et al., 2015; Deng et al., 2017; Fang et al., 2017). Therefore
the discussion on SOAFPs in this study is only limited to traditional anthropogenic
SOA precursor species (mainly aromatic hydrocarbons), and intermediate-volatility
organic compounds (IVOCs), which is a large of secondary organic aerosol (Zhao et
al., 2014), should be further considered in order to fully understand the influence of
control measures on the ambient SOAs.
**4. Conclusions**
During severe wintertime haze events in recent years in Beijing, SOA often
shared higher factions in organic aerosols, yet their precursor VOCs in ambient air
during winter are much less understood especially in the rural areas. In this study we
collected ambient air samples from 25 October to 31 December in 2014 at a rural site
inside the campus of UCAS in north Beijing. As the APEC summit was held in
Beijing during 5-11 November 2014 and temporary control measures were adopted to
improve the air quality and in fact the so-called "APEC Blue" was achieved due to the



enhanced emission control. Therefore we could take advantage of this opportunity to
see how the control measures influence the ambient VOCs in the rural areas. On the
other hand, wintertime heating with coal burning has been regarded as major
contributor to wintertime PM pollution and haze events, thus we could also compare
the ambient VOC levels and compositions before and after the start of central heating
on 15 November, and investigate the influence of central heating on ambient VOCs
based on our observation at the rural site.

We observed that during the enhanced emission control period II (3-12

November) average mixing ratios of VOCs decreased ~50% when compared to that
before or after that period. And their ozone and SOA formation potentials accordingly
decreased by ~50% and ~70%, respectively as a result of temporary intervention
measures implemented during period II. The larger drop in SOA formation potentials
was attributable to more effective control of aromatic hydrocarbons mainly from
solvent use. Based on PMF source apportioning, the control of traffic-related
emissions (gasoline and diesel exhaust) and solvent use could explain 47.9 and 37.6%
of the reduction in ambient VOCs. This result thus offered an observation-based
evaluation about the temporary emission control measures.

With back trajectory analysis, we could compare ambient VOCs with the change

of wind directions and thus further investigate source emission strength in different
regions. Total mixing ratios of VOCs in the southerly air masses were 2.3, 2.3 and 2.9
times that in the northerly air masses before, during and after the period with
temporary emission control for the APEC summit. VOC episodes during the



campaign all occurred under southerly winds. This confirms that emission control in
the southern urbanized regions is crucial for reducing ambient VOCs.

As residential coal/biomass burning were not controlled during the APEC

summit, its contribution to the ambient mixing ratios of VOCs was similar between
period I and period II, although contribution percentages by coal/biomass burning
became the largest in average due to drops in the percentages by other sources. During
period III with central heating, coal/biomass burning became the largest source that
accounted for 45.1% of the VOCs. Specifically, during period III coal/biomass
combustion contributed 38.2% of VOCs in the southerly air masses (or in the south
regions), and 48.8% of VOCs in the northerly air masses (or in the north regions).

The finding of this study will provide useful information on the direction of

control strategies of VOCs for abating both ozone and $PM_{2.5}$ pollution. The reduction
in total OFPs and SOAFPs during the APEC is largely due to the drop of reactive
alkenes and aromatics, so adopting reactivity-based emission control would be the
effective and economical way to lower the ozone and SOA formation potentials of
VOCs. As control measures related to solvent use and vehicle exhausts explained
most of the reduction in both ambient VOCs and their ozone/SOA formation
potentials, enhancing emission control for solvent use (especially solvents with
aromatic hydrocarbons) and vehicle exhaust would benefit improving air quality in
the future. Moreover, as observed in this study, even in megacities like Beijing,
burning raw coal or biomass for household heating in winter could contribute near
half of VOCs in ambient air, therefore a cleaner way of wintertime household heating





would help to lower both primary emission and secondary formation of air pollutants.

**Acknowledgments**
This study was supported by the National Key Research and Development Program
(2016YFC0202204/2017YFC0212802), the Chinese Academy of Sciences (Grant No.
QYZDJ-SSW-DQC032), the National Natural Science Foundation of China (Grant
No. 41673116/41530641/41571130031), and the Guangzhou Science Technology and
Innovation Commission (201607020002/201704020135).



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



Table 1. Enhanced temporary air pollution control measures during the 2014
Asian-Pacific Economic Cooperation (APEC) summit

| Control types | Details | Control areas |
|---|---|---|
| Traffic | 1. Yellow label vehicles were banned to run inside the sixth ring of Beijing and the Huairou urban area; 2. The number of private vehicles in operation reduced by 50% through an odd/even-number-plate rule throughout Beijing; 3. Trucks were limited to drive inside the sixth ring of Beijing between 6 am and 24 pm; 4. 70% of government cars also ordered off the roads in Beijing. | Inside the sixth ring of Beijing and the Huairou urban area |
| Industrial | 1. 9289 enterprises were suspended, 3900 enterprises were ordered to limit production; 2. More than 40000 construction sites were shut down. | Beijing, some areas of Tianjin, Hebei, Shanxi, Shandong and Inner Mongolia |
| Other fields | 1. Open fire was completely controlled at North China Plain; 2. Increasing road cleaning and water spraying in Beijing; 3. Other relate control measures carried out in surrounding areas. | North China Plain Tianjin, Hebei, Shanxi, Shandong and Inner Mongolia |






Table 2. The mixing ratios, ranges and 95% confidence intervals (95% C.I.) of VOCs
during period I, II and III at the rural site inside UCAS (in parts per trillion by volume,
pptv).

| Species | MDL[a] | Period I | | Period II | | Period III | |
|---|---|---|---|---|---|---|---|
| | | Range | Mean (95% C.I.) | Range | Mean (95% C.I.) | Range | Mean (95% C.I.) |
| Ethane | 39 | 1172-7855 | 3254(743) | 910-5511 | 2442(491) | 1082-12714 | 3674(465) |
| Propane | 31 | 427-6145 | 2880(720) | 270-4138 | 1296(384) | 598-7604 | 2479(329) |
| i-Butane | 14 | 53-2755 | 1121(312) | 59-1400 | 474(187) | 106-2741 | 754(129) |
| n-Butane | 21 | 158-2947 | 1283(302) | 83-1735 | 562(196) | 174-3047 | 841(136) |
| i-Pentane | 14 | 94-3729 | 1425(354) | 39-1388 | 561(167) | 72-12590 | 1018(279) |
| n-Pentane | 8 | 47-1697 | 615(182) | 30-910 | 247(106) | 26-4808 | 456(112) |
| 2,2-Dimethylbutane | 14 | 15-68 | 30(6) | 17-32 | 24(3) | BDL[b]-75 | 26(2) |
| Cyclopentane | 12 | 13-135 | 64(15) | BDL-64 | 35(7) | 13-274 | 50(8) |
| 2,3-Dimethylbutane | 12 | 13-140 | 45(15) | 22-51 | 32(4) | 13-235 | 38(6) |
| 2-Methylpentane | 8 | 13-679 | 171(68) | 12-257 | 77(31) | 9-1077 | 124(27) |
| 3-Methylpentane | 7 | 12-548 | 150(54) | 14-220 | 68(26) | 16-792 | 104(20) |
| n-Hexane | 6 | 115-1033 | 505(97) | 102-921 | 324(89) | 108-7393 | 1400(257) |
| Methylcyclopentane | 9 | 10-283 | 100(30) | 13-195 | 59(23) | BDL-535 | 88(17) |
| 2,4-Dimethylpentane | 4 | BDL-43 | 15(5) | BDL-15 | 10(2) | BDL-90 | 16(2) |
| Cyclohexane | 6 | 10-458 | 167(51) | 10-107 | 43(14) | 7-646 | 76(17) |
| 2-Methylhexane | 6 | 10-304 | 68(27) | 7-56 | 22(6) | 7-318 | 51(10) |
| 2,3-Dimethylpentane | 9 | BDL-139 | 31(12) | BDL-24 | 15(2) | BDL-102 | 28(3) |
| 3-Methylhexane | 6 | 12-436 | 93(38) | 8-97 | 41(11) | 9-367 | 70(12) |
| 2,2,4-Trimethylpentane | 9 | 12-126 | 44(12) | BDL-41 | 25(4) | BDL-127 | 38(5) |
| n-Heptane | 10 | 12-358 | 89(33) | 12-71 | 30(8) | 13-441 | 82(14) |
| Methylcyclohexane | 5 | BDL-162 | 51(17) | BDL-66 | 21(7) | BDL-162 | 44(8) |
| 2,3,4-Trimethylpentane | 6 | BDL-38 | 14(4) | BDL-12 | 9(1) | BDL-59 | 16(2) |
| 2-Methylheptane | 4 | 8-175 | 31(16) | BDL-31 | 13(3) | BDL-91 | 22(3) |
| 3-Methylheptane | 5 | BDL-231 | 26(20) | BDL-15 | 8(1) | BDL-74 | 17(2) |
| n-Octane | 6 | 8-104 | 42(11) | BDL-31 | 18(3) | BDL-160 | 40(6) |
| n-Nonane | 6 | 9-99 | 40(11) | BDL-37 | 18(4) | BDL-171 | 38(6) |
| n-Decane | 6 | 14-777 | 129(74) | 8-110 | 36(14) | BDL-600 | 73(17) |
| n-Undecane | 7 | 47-317 | 151(35) | 27-206 | 66(20) | 11-374 | 94(12) |
| n-Dodecane | 7 | 9-646 | 129(57) | 25-313 | 75(30) | 8-316 | 63(9) |
| Ethylene | 41 | 367-3495 | 1788(391) | 553-3572 | 1254(352) | 319-13911 | 2313(428) |
| Propene | 31 | 117-1264 | 430(118) | 170-766 | 371(67) | 176-3222 | 820(128) |
| 1-Butene | 17 | 19-161 | 107(18) | BDL-100 | 55(12) | 19-581 | 137(22) |
| 1,3-Butadiene | 20 | 21-403 | 154(44) | 23-234 | 79(27) | BDL-2140 | 252(74) |
| trans-2-Butene | 5 | BDL-41 | 18(4) | BDL-35 | 12(4) | BDL-425 | 39(10) |
| cis-2-Butene | 7 | 9-50 | 23(4) | BDL-38 | 14(5) | BDL-276 | 37(7) |
| 1-Pentene | 20 | BDL-47 | 33(3) | 21-25 | 23(1) | BDL-127 | 52(6) |
| Isoprene | 13 | BDL-623 | 163(56) | 16-143 | 62(15) | 17-765 | 200(24) |
| trans-2-Pentene | 10 | BDL-37 | 17(4) | BDL-19 | 15(3) | BDL-65 | 23(3) |
| cis-2-Pentene | 6 | BDL-24 | 11(3) | BDL-9 | 8(0) | BDL-46 | 15(2) |
| 2-Methyl-2-butene | 12 | 13-50 | 21(4) | 17-20 | 18(1) | BDL-61 | 24(2) |
| Benzene | 14 | 75-2695 | 868(279) | 43-1465 | 410(179) | 72-2916 | 795(151) |
| Toluene | 9 | 120-3585 | 1273(419) | 47-1186 | 343(126) | 62-3425 | 840(146) |
| Ethylbenzene | 6 | 25-2210 | 684(240) | 12-611 | 145(67) | 23-2450 | 317(75) |
| m/p-Xylene | 9 | 39-2106 | 765(248) | 16-620 | 149(67) | 25-3285 | 422(91) |
| Styrene | 8 | 15-578 | 167(71) | BDL-99 | 32(11) | 10-1267 | 151(38) |
| o-Xylene | 4 | 11-965 | 334(104) | 9-284 | 71(31) | 15-1224 | 178(36) |
| Isopropylbenzene | 4 | 5-66 | 24(7) | BDL-21 | 11(2) | BDL-77 | 18(3) |
| n-Propylbenzene | 4 | 6-231 | 71(27) | BDL-55 | 20(7) | 5-239 | 38(8) |
| m-Ethyltoluene | 3 | 13-593 | 136(67) | 4-91 | 28(11) | 4-854 | 85(23) |
| p-Ethyltoluene | 3 | 6-295 | 61(29) | 4-59 | 17(6) | 4-245 | 41(9) |
| 1,3,5-Trimethylbenzene | 3 | 7-217 | 48(21) | BDL-35 | 12(4) | 4-179 | 38(6) |
| o-Ethyltoluene | 3 | 5-246 | 64(26) | 4-58 | 17(6) | 5-230 | 40(8) |
| 1,2,4-Trimethylbenzene | 6 | 22-984 | 220(93) | 13-219 | 58(22) | 8-803 | 127(26) |
| 1,2,3-Trimethylbenzene | 5 | 12-442 | 82(37) | BDL-92 | 24(9) | 6-300 | 56(11) |
| 1,3-Diethylbenzene | 4 | 11-135 | 35(12) | BDL-26 | 15(3) | BDL-126 | 26(4) |
| 1,4-Diethylbenzene | 4 | 14-461 | 80(40) | 5-69 | 23(7) | BDL-292 | 51(10) |
| 1,2-Diethylbenzene | 4 | BDL-30 | 15(4) | BDL-8 | 6(1) | BDL-76 | 15(2) |
| Ethyne | 57 | 406-10539 | 3128(1043) | 290-6260 | 1625(615) | 584-10378 | 3008(509) |

[a] MDL, method detection limits, pptv; [b] BDL, bellowed detection limit.



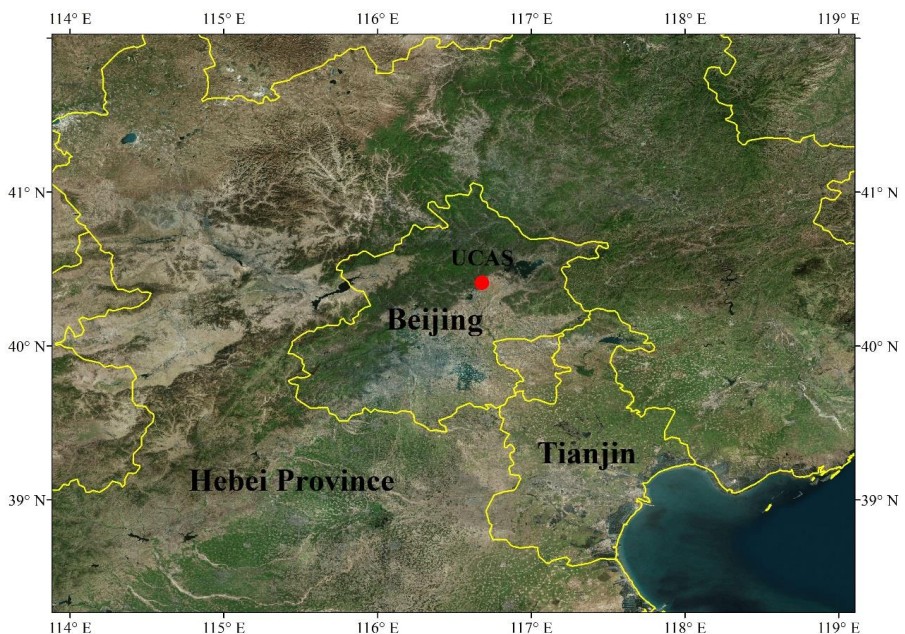


Figure1. Location of sampling site at a rural inside the campus of University of
Chinese Academy of Science (UCAS).



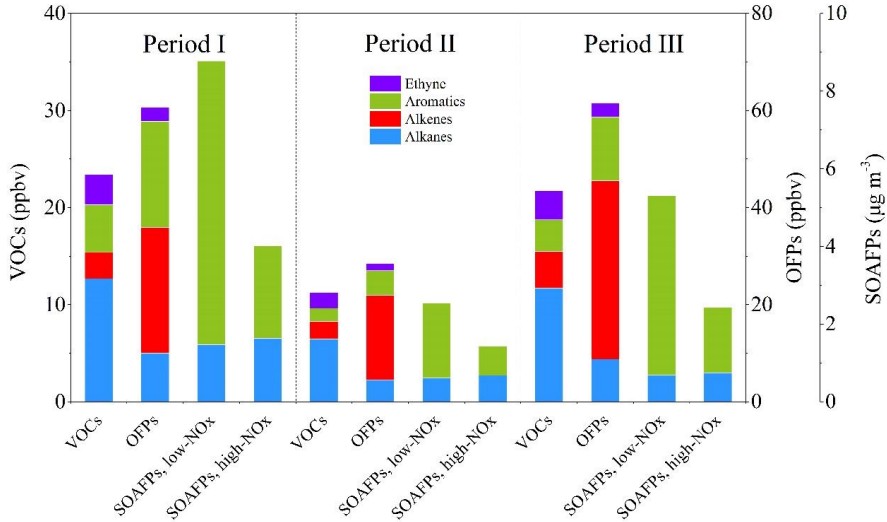


Figure 2. Mixing ratios of VOCs, ozone formation potentials (OFPs) and secondary
organic aerosol formation potentials (SOAFPs) during period I, II and III at UCAS,
respectively.





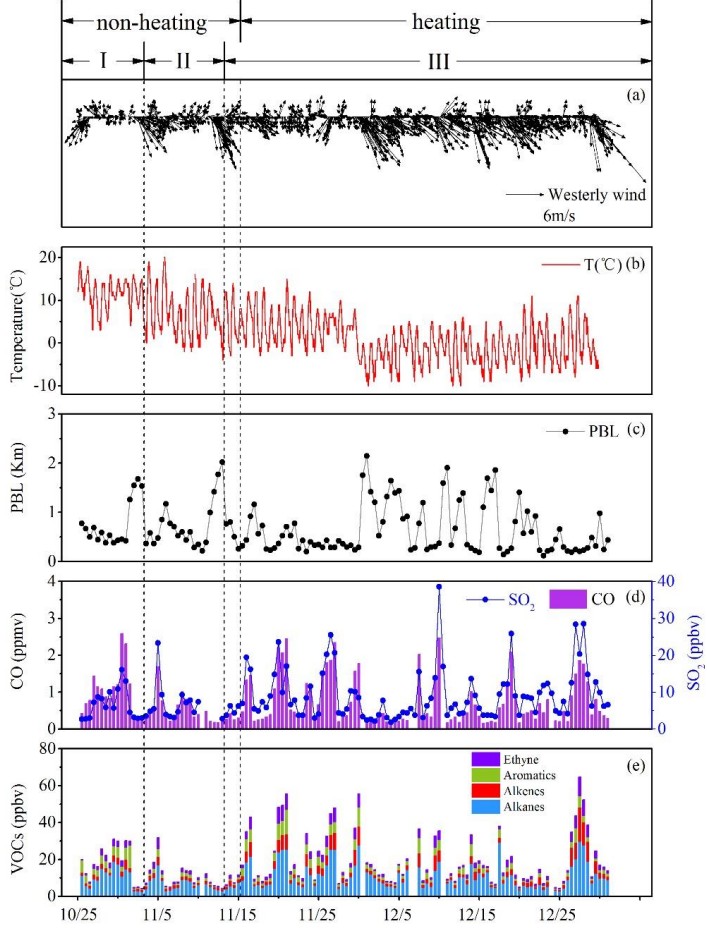


Figure 3. Time series of (a) wind speed and wind direction, (b) temperature, (c)
planetary boundary layer height, (d) mixing ratios of CO and SO₂, (e) mixing ratios of
VOCs, at the sampling site inside UCAS. The heating periods started on 15 November.
Period I: 25 October-2 November; Period II: 3-12 November; Period III: 13
November-31 December.



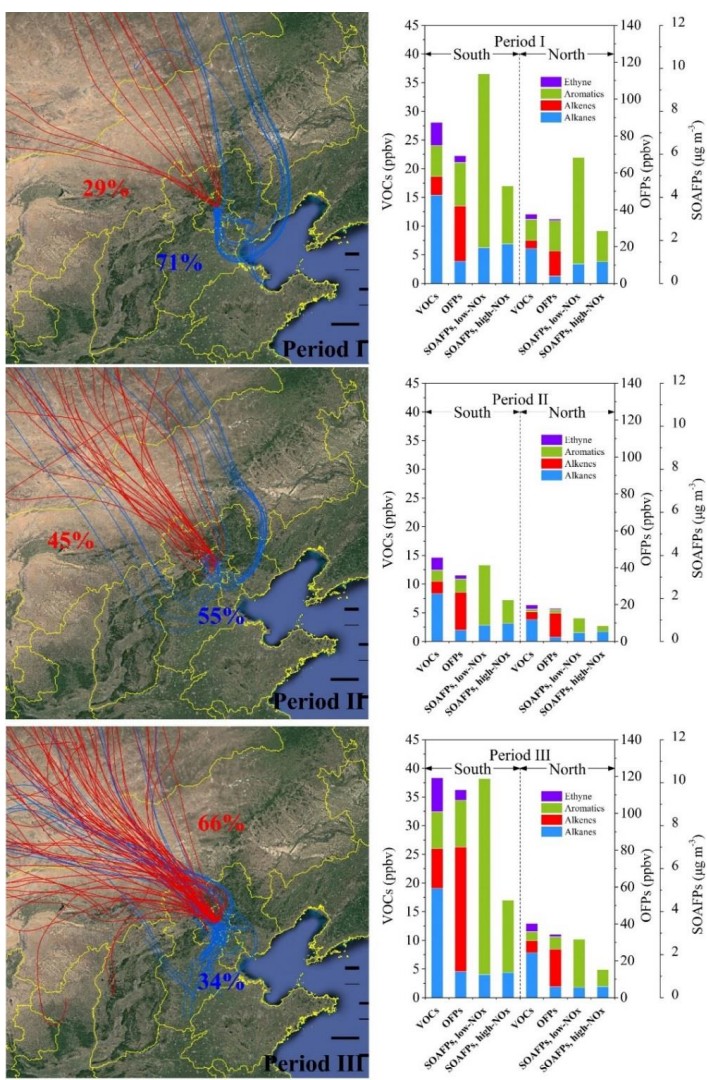


Figure 4. Mixing ratios of VOCs, ozone formation potentials (OFPs) and secondary
organic aerosol formation potentials (SOAFPs) in the air masses from the south and
north regions (right) and corresponding back trajectories at 100 meters above the
ground level during period I, II and III, respectively (Left).





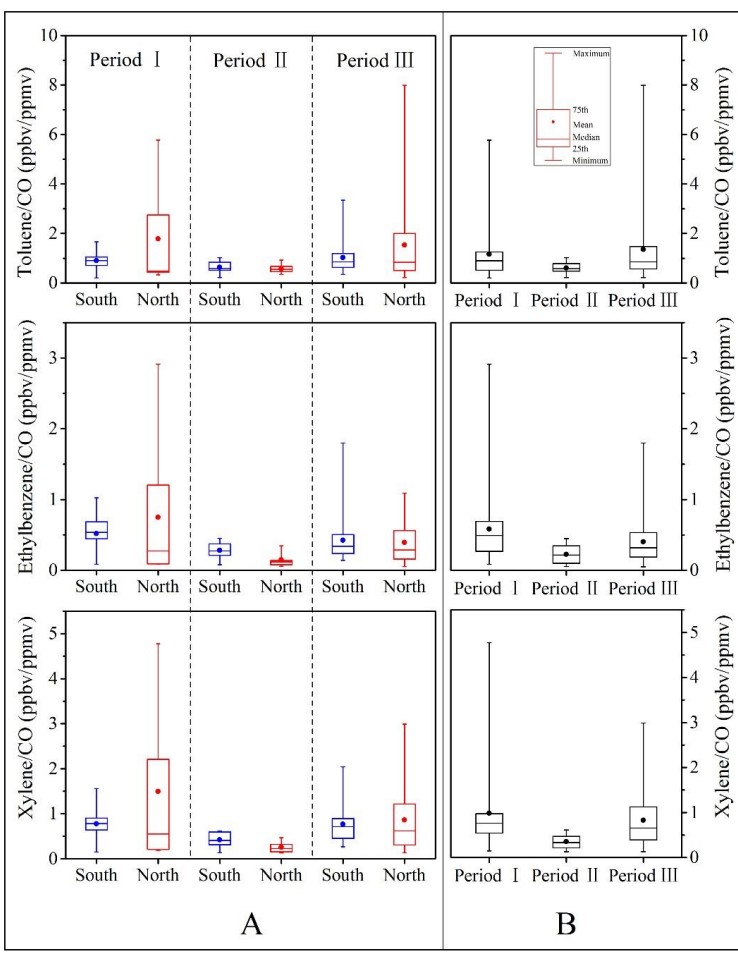


Figure 5. Ratios of aromatic hydrocarbons to carbon monoxide (CO) (A) in the air

masses from the south and north regions and (B) in all samples during period I, II and

III. (The lower and upper boundaries of the box represent the 25th and 75th

percentiles, respectively; the whiskers below and above the box indicate the minimum

and maximum, respectively; the line within the box marks the median; the dot

represent the mean).






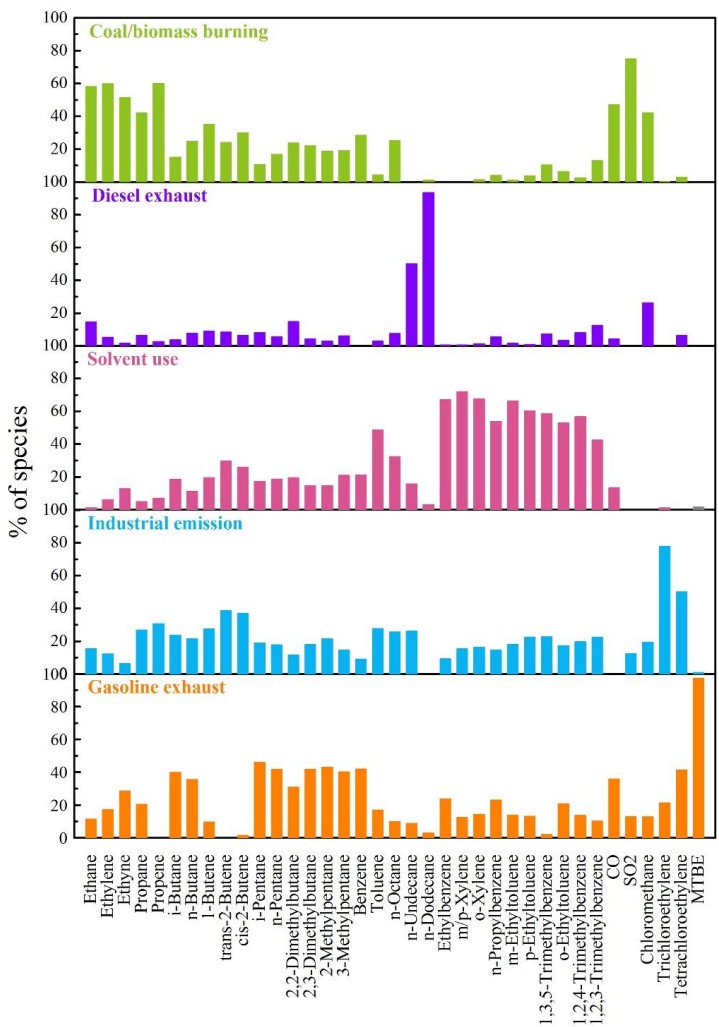


Figure 6. Source profiles revolved by PMF.





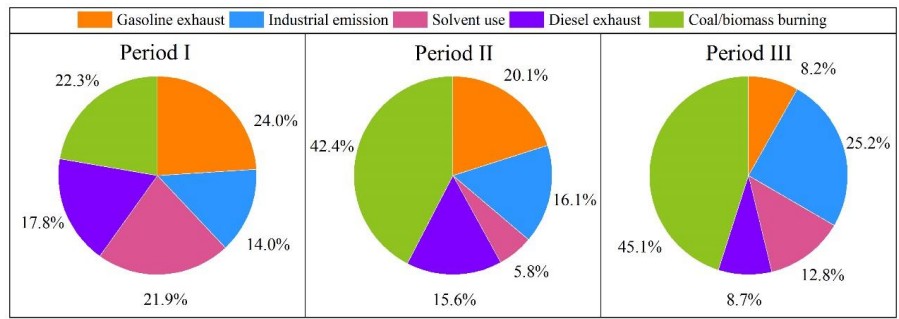


Figure 7. Contributions to VOCs in percentages (%) by different sources during
period I, II and III.





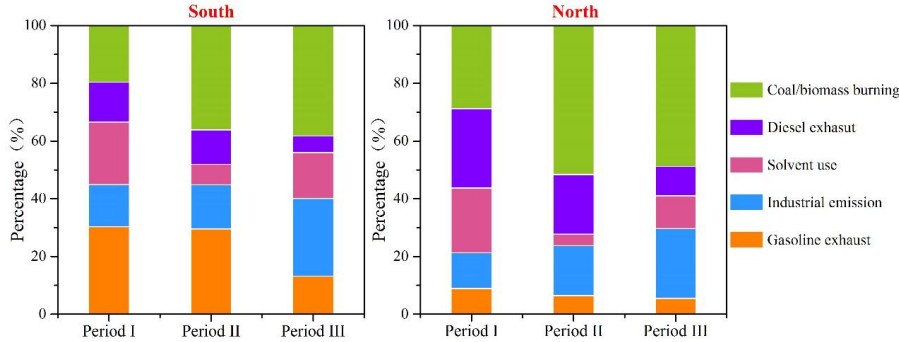

Figure 8. Sources contributions (%) to VOCs in the air masses from the south and north regions during period I, II and III.



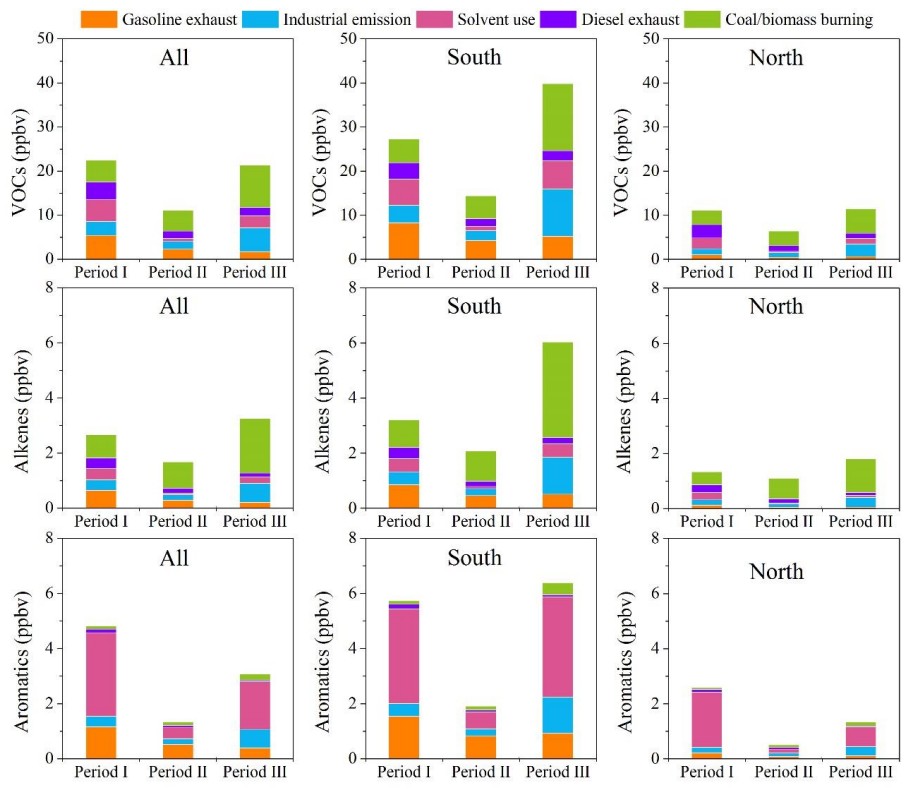


Figure 9. Sources contributions of VOCs and reactive alkenes/aromatics at UCAS, in
all samples and in air masses from the south and north regions during period I, II and
III.





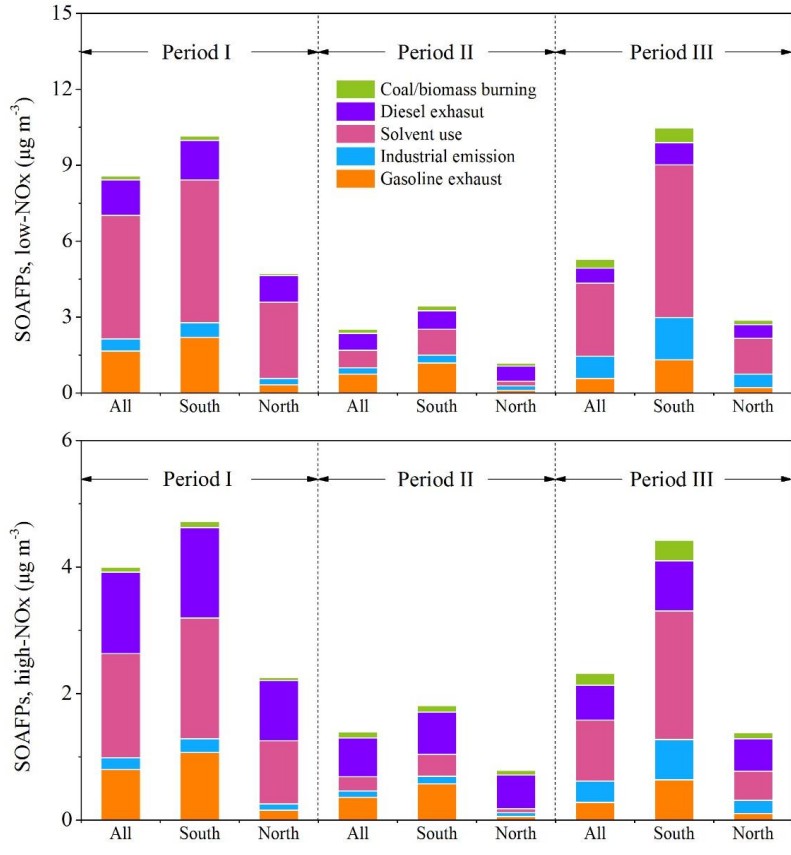


Figure 10. Contributions to SOAFPs by different sources in the air masses from the
south and north regions during period I, II and III.