# Peer review of "Volatile organic compounds at a rural site in Beijing: Influence of temporary emission control and wintertime heating"

_Atmospheric Chemistry and Physics, 2018_

## Referee Comment (RC1) · Anonymous Referee #1 · 29 Apr 2018

Yang et al. present VOC observational results at a rural site in the Beijing metropolitan area before, during, and after a strict emission control implementation for the APEC summit event. They mainly discussed on changes in the VOC composition during three different periods. Then, they move ahead to discuss the potential sources from the PFM analysis and the impacts of VOCs in the context of ozone and aerosol formation rates. Overall the manuscript is very clearly written, and the goals seem to be well achieved.

However, I have a couple of concerns regarding whether this manuscript is well fit in this particular journal. Although the significance in understanding of Chinese pol-

lution problems cannot be overstated considering the large population in China, this manuscript is too narrowly focused on Chinese local air pollution problems without discussion on the bigger context such as regional or global impacts. Moreover, the scientifically relevant analyses such as ozone forming potential and secondary aerosol forming potential are just adapted from previous publications without detailed discussion whether the method is relevant to this particular photochemical environment. I believe that ACP aims more detailed scientific discussion relevant to the broader atmospheric research community, but this manuscript too narrowly discusses the scientific implications of the dataset. I recommend expanding discussion at least to compare VOC speciation from other locations in the region and other metropolitan areas in the world. In addition, I would recommend more thorough descriptions on ozone formation potential and SOA formation potential particularly in the context of whether the metrics are relevant in this photochemical environment.

---

## Referee Comment (RC2) · Anonymous Referee #3 · 5 Jun 2018

Household air pollution from burning biomass and coal for cooking foods and heating rooms has long been a major environmental problem. Previous studies suggested that the uncontrolled and inefficient combustion of solid fuels for heating in winter also contributed substantially to outdoor PM2.5, BC, OC, SO2 and NOx in many regions, such as in the North China Plain (Liu et al., 2016, PNAS). The field observations at the rural site of Beijing in this study further demonstrated that the combustion of solid fuels for heating in winter made remarkable contribution to ambient volatile organic compounds (VOCs). The authors also took advantage of the temporary intervention measures for emission control during the APEC to evaluate the actual effect of the control measures on the ambient VOCs levels through reduction of the source contributions. The comparison between the heating and non-heating periods offered robust results indicating the influential emission from winter heating. Overall, the data quality of this manuscript is quite good, and the interpretation of the results is appropriate and convincing. Therefore, I strongly recommend publication of this manuscript. Majors: As stated above, an important finding of this study is that residential coal burning, especially during winter heating period, could be a major contributor to the ambient VOCs. Apart from the emission factors available for residential coal burning, is it possible for the authors to have the amounts of coals consumed in the residential sector especially during winter and give an in-depth explanation why this source could contribute substantially? The study suggests clean energy use in residential sector is vital for reducing VOCs in the heavily polluted winter. This aspect should be incorporated into the discussion or the conclusion part of the manuscript.

Minors: Although I am not a native English speaker, I would say that there is still room for improving English writing of the manuscript. The authors better find a native English speaker to check the English. Below is a list of changes that, I think, should be made: Line 33-34: Change "during wintertime severe haze events" to "during severe wintertime haze events "; Line 35: Change "comparatively much less" to "not well"; Line 37: Change "inside" to "on" Line 37: add "the" before "University"; Line 38: Change "northeast" to "northeastern"; Line 39: Remove "that"; Line 39: Change "during" to "on"; Line 40: Add "the" before "air quality"; Line 41: Change "in" to "on"; Line 41: Remove "that"; Line 41: Change "since" to "on"; Line 42: Change "it is" to "this sample collection period provided"; Line 43: Remove "the" before "temporary" and before "wintertime"; Line 44, 45: Add "the" before "temporary"; Line 46: Change "about" to "approximately"; Line 47: Change "that" to "the values"; Line 46-48: Change "that of 23.41 ppb before the APEC (25 October-2 November; Period I) or 21.71 ppb after the APEC (13 November-31 December; Period III)" to "the values of 23.41 ppb in Period I (25 October-2 November) before the APEC and 21.71 ppb in period III (13 November-31 December) after the APEC"; Line 48: Change "Their" to "The"; Line 49: Change "drop" to "decrease"; Line 50: Change "of" to "over"; Line 51: Remove "the" before "southerly"; Line 52: Change

"that in the northerly ones during period I, II and III" to "those in northerly air masses during periods I, II and III"; Line 53: Remove "and"; Line 54: Change "south" to "southern"; Line 56: Change "changed" to "the altered"; Line 57, 58: Remove "the" before "Period"; Line 58: Change "emission" to "emissions"; Line 59: Add "the" before "ambient"; Line 60: Change "emission" to "emissions"; Line 60: Remove "they"; Line 61-62: Change "VOCs during the period I, II and III" to "the VOCs during periods I, II and III"; Line 62-63: Change "became the dominant source which accounted for 45.1% of the VOCs" to "became the largest source, accounting for 45.1% of the VOCs"; Line 64-65: Change "with a remarkably lower average contribution percentage (38.2%) in the southerly air masses than that of 48.8% in the northerly air masses" to "with a specifically lower average contribution percentage in southerly air masses (38.2%) than in northerly air masses (48.8%)"; Line 72: Change "emission" to "emissions"; Line 73: Change "benefit improving" to "improve the"; Line 74: Change "an extensive concern" to "a widespread concern"; Line 75, 79: Change "summertime" to "summer"; Line 75: Change "wintertime" to "winter"; Line 80: Remove "as well"; Line 82: Change "comparatively the role of VOCs in the wintertime with PM2.5" to "comparatively, the effect of VOCs on wintertime PM2.5"; Line 88: Change "in the control of air pollution by PM2.5 in wintertime" to "in the control of PM2.5 air pollution in winter"; Line 90: Change "under" to "on"; Line 92: Change "vehicle exhausts are" to "vehicle exhaust is an"; Line 96: Change "particularly in north China in wintertime" to "particularly in northern China in winter"; Line 99: Add "the" before "VOCs"; Line 100: Change "is widely occurring in the rural areas" to "widely occur in rural areas"; Line 101-102: Change "how the enhanced emission" to "how enhanced emissions"; Line 109: Change "north" to "northern"; Line 110: Add "the" before "air quality"; Line 111: Remove "so"; Line 111: Remove "when"; Line 112: Change "especially in wintertime with" to "especially in winter, with"; Line 116: Change "kind" to "type"; Line 118: Change "As for" to "For"; Line 121: Change "drops of" to "decreases in"; Line 123: Change "inside" to "on"; Line 124: Add "the" before "total"; Line 124: Add "were" before "reduced"; Line 125: Add "those in" before "the period"; Line 126: Change "about" to "of"; Line 127, 134: Remove "the" before

"urban"; Line 127: Remove "entirely"; Line 128: Add "on" before "a regional scale"; Line 132-133: Change "The objectives of present study are" to "The objectives of the present study are as follows"; Line 135, 136: change "crucial" to "the major"; Line 136: Change "wintertime" to "winter"; Line 137: Add "the" before "APEC"; Line 141: Change "○" to "°"; Line 142: Change "inside the campus of" to "on the campus of the"; Line 142: Add "the" before "Huairou"; Line 143-144: Change "The UCAS is located about 60 km northeast of central Beijing and about 150 km northwest of the Tianjin city" to "UCAS is located approximately 60 km northeast of the center of Beijing and approximately 150 km northwest of the city of Tianjin"; Line 145-147: Change "16 meters above ground on the top of a four-story building, about 100 m west of a national road and only 1.5 km far away from the APEC main conference hall" to "16 meters above the ground on the top of a four-story building, approximately 100 m west of a national road and only 1.5 km away from the main APEC conference hall"; Line 152: Remove "of"; Line 153: Add "and" before "one"; Line 154: Add "was" before "less than"; Line 154: Add "a" before "relative"; Line 154: Add "of" before "less"; Line 158: Remove "the time span"; Line 163-164: Change "and average temperature was 11.4, 7.0, and 0.6ïĆřC during periods I, II and III" to "and the average temperature was 11.4, 7.0 and 0.6°C during periods I, II and III"; Line 224: Remove "the" before "period"; Line 226: Change "Total" to "The total"; Line 226: Change "inside" to "at"; Line 227: Change "in" to "on"; Line 229: Change "halves of 57.45, 36.17, and 56.56 ppb" to "half the values (57.45, 36.17, and 56.56 ppb)"; Line 231: Remove "both"; Line 234: Add "a" before "more than"; Line 234: Remove "the"; Line 234: Change "about" to "an approximately"; Line 237: Change "or" to "and the"; Line 238: Change "densities" to "density"; Line 240: Change "shared by" to "of"; Line 240: Remove "quite"; Line 243-244: Change "percentages shared by aromatics became lower during period II (12%) when compared to that in period I (21%) or period III (15%)" to "the percentage of aromatics was lower during period II (12%) than during period I (21%) and period III (15%)"; Line 246-247: Change "decreased by 49.0, 32.5, 72.8, and 48.1%, respectively, when compared to those during period I" to "were 49.0, 32.5, 72.8 and 48.1% lower than those during period

I, respectively"; Line 247-248: Change "Aromatics evidently had a more substantial drop" to "Aromatics evidently underwent a larger decrease"; Line 249: Remove "as"; Line 252: Change "ozone formation potentials (OFPs)" to "ozone formation potential (OFP)"; Line 253: Change "in average during periods I, II and III were" to "on average during periods I, II and III was"; Line 255: Change "Their" to "The"; Line 255: Change "potentials (SOAFPs)" to "potential (SOAFP)"; Line 257, 259: Add "the" to "total"; Line 261: Change "VOCs" to "VOC"; Line 262: Change "drop" to "decreases"; Line 264: Change "changed contribution by aromatic" to "the altered contribution of aromatics"; Line 266: Change "condition" to "conditions and from"; Line 267: Change "condition" to "conditions"; Line 270: Change "showed" to "shown"; Line 271: Change "with the increase in" to "with an increase in the"; Line 272: Change "like that" to "as those"; Line 276: Add "the" before "wind"; Line 280: Change "showed" to "shows"; Line 281: Change "the" to "a"; Lines 282-285: "It clearly demonstrated that the mixing ratios of VOCs increased rapidly, and the back trajectories indicated that air masses changed from northerly to southerly and then declined sharply while the air masses changed back from southerly to northerly", rewrite the sentences. Lines 285-286: Change "The southern areas of UCAS are the central Beijing with stronger emissions" to "The southern areas of UCAS are in central Beijing where emissions are stronger"; Line 286: Add "that"; Line 287: Change "increase of" to "increase in the"; Line 288: Change "of source regions" to "in the source region"; Line 289: Change "showed" to "shown"; Line 291: Change "changed" to "altered". Line 295: Add "and" before "2)"; Line 296: should be "…Mongolia and quickly…"; Line 297: Change "emission" to "emissions"; Line 302: Change "period" to "periods"; Line 303: Change "that" to "those"; Line 304: Change "OFPs in the southerly air masses were" to "the OFP in southerly air masses was"; Line 305: Change "and SOAFPs in the" to "and the SOAFP in"; Line 305: Change "were" to "was"; Line 308: Change "This indicates that the north and south" to "These results indicate that the northern and southern"; Line 313-314: Change "OFPs and SOAPFs" to "OFP and SOAFP"; Line 314: Add "that" before "the changes"; Line 315: Remove "to"; Line 316: Change "when compared to that during period I average" to "compared to

that during period I, the average"; Line 319: Change "OFPs decreased by 48.1% and SOAFPs" to "the OFP decreased by 48.1% and the SOAFP"; Line 320: Change "when compared to that" to "compared to those"; Line 321: Add "the" before "average"; Line 323: Remove "when"; 144. Line 323-324: Change "OFPs decreased by 48.9% and SOAFPs decreased by over 70% during period II relative to period I" to "the OFP decreased by 48.9%, and the SOAFP decreased by over 70% during period II relative to those in period I"; Line 325: Add "a" before "more"; Line 326: Change "emission control in" to "control over emissions from"; Line 327: Change "less changes in mixing ratios" to "decreased changes in the mixing ratios"; Line 328: Change "of" to "over"; Line 331: Change "that" to "those"; Line 331: Change "This difference in the increase rates" to "These different increases"; Line 332: Change "are" to "were"; Line 333: Change "heating supply was only available since" to "heat sources were only available after"; Line 334: Change "already" to "have"; Line 336: Change "apportioning" to "apportionment"; Line 338: Add "the" before "campaign"; Lines 338-339: Change "might be resulted from changed contribution by emission sources, such like" to "might have resulted from the altered contributions from emission sources, such as"; Line 341: add "the" before "characteristic" Line 344: Change "in average during period" to "on average during periods"; Line 345: Change "were approaching" to "approached"; Line 347, 351: Remove "the" before "period"; Line 348: Remove ", which is characteristic of vehicular exhaust"; Line 349: Add "the" before "incomplete": Line 354: Change "in" to "on"; Line 355: Change "that coal burning contributed more" to "an increased contribution of coal burning"; Line 357: Change "emission" to "emissions"; Line 359: Change "are mainly from" to "mainly originate from"; Line 361: Change "were" to "are"; Line 364: Change "when compared to that of" to "than the values of"; Line 365: Change "or" to "and of"; Line 366: Change "This drop in aromatics/CO ratios" to "This decrease in the ratios of aromatics to CO"; Line 367: Change "of" to "over"; Line 370, 371: Add "those in" before "period I"; Line 371-372: Change "Apparently larger decrease in TEX/CO ratios in the northerly air masses reflected" to "A larger decrease in the TEX/CO ratios in northerly air masses reflects the fact that"; Line 374: Change "3.3.2 Source Apportioning by PMF" to "3.3.2

Source Apportionment by PMF"; Line 375: Change "Thirty-five" to "The 35"; Line 375: Add "and" before "ethyne"; Line 376: Change "sources" to "source"; Line 377: Add "use with" before "the PMF"; Line 385: Change "the gasoline vehicle emission" to "gasoline vehicle emissions"; Line 386: should be "trichloroethylene and tetrachloroethylene"; Line 387: Change "of" to "by"; Line 388: Change "manufacturing industrials" to "industrial manufacturing"; Line 389-390: Change "by industries for make" to "in industry to prepare"; Line 390: Add "during the" before "production"; Line 392: Change "emission" to "emissions"; Line 393: Change "a larger percentage of the" to "larger percentages of"; Line 394: Chang "It is known that TEX are" to "TEX is known to be"; Line 396: Change "They are also" to "These compounds are also the"; Line 396-397: Change "auto factory painting and building coating" to "automobile factories, paint and building coatings"; Line 398: Change "as" to "to be"; Line 403-404: Change "top 3 species of" to "the top 3 species emitted during"; Line 405: Change "were" to "are"; Line 405-406: Change "and aromatics like benzene" to "as well as aromatics such as benzene"; Line 406-407: Change "So factor 5 is related to the coal/biomass burning" to "Thus, factor 5 is related to coal/biomass burning"; Line 408: Change "period" to "periods"; Line 409: Add "the" before "VOCs"; Line 411: add the "the" before "temporary" and remove "the" before "period"; Line 412: Change "by" to "of"; Line 413: Change "drop" to "decrease"; Line 414: Change "by" to "from"; Line 415: Change "Quite similar" to "Similar"; Line 417: Remove "the"; Line 418: Change "from" to "on"; Line 419: Change "emission" to "emissions"; Line 420: Add "the" before "VOCs"; Line 421: Change "were showed" to "are shown"; Line 422: Change "by" to "of"; Line 422: Change "with the" to "with the increase in"; Line 423: Remove "instead"; Line 427: Change "consumptions" to "consumption"; Line 429: Change "The residential coal combustion is prevailing" to "Residential coal combustion is primarily carried out"; Line 429: Change "by using" to "with"; Line 430: Add "the" before "rural"; Line 431: Change "wintertime" to "winter"; Line 431: Remove "the"; Line 431: Change "accounts" to "accounted"; Line 433: Change "contribute predominately to" to "have been the predominant contributor to the"; Line 437, 438, 444, 446, 449: Remove "the" before "period"; Line 437, 440, 441: Change "by"

to "from"; Line 441: Change "reduced" to "decreased"; Line 443: Change "showed" to "made"; Line 443: Change "period" to "periods"; Line 447: Change "account" to "accounted"; Line 451: Change "period" to "periods"; Line 451: Remove "respectively"; Line 452: Change "emission (gasoline and diesel vehicles) was" to "emissions (gasoline + diesel) were"; Line 453: Change "VOCs during the period" to "the VOCs during periods"; Line 454: Remove "instead"; Line 456: Change "period" to "periods"; Line 459: Change "Contributions" to "The contributions"; Line 460: Change "were" to "are"; Line 460-461: Change "was mainly coming" to "mainly originated"; Line 467, 468, 470: Change "of" to "in"; Line 472: Change "emission" to "emissions"; Line 474: Change "3.3.3 Source contributions to SOAFPs" to "3.3.3 Source contributions to the SOAFP"; Line 475: Change "apportioning" to "apportionment"; Line 475: Change "SOAFPs by" to "the SOAFP of"; Line 476: Change "showed" to "shown"; Line 477: Change "condition" to "conditions"; Line 477: Change "SOAFPs by solvent use were much higher than that by" to "SOAFP of solvent use was much higher than that of"; Line 478: Change "were" to "was"; Line 479, 481: Change "period" to "periods"; Line 482: Change "of SOAFPs" to "in SOAFP"; Line 483: Change "by" to "from"; Line 485: Change "condition," to "conditions, the"; Line 485: Change "of" to "in"; Line 486: Change "by" to "from"; Line 490: Change "or" to "and"; Line 490: Change "from" to "during"; Line 494: Change "SOAFPs" to "the SOAFP"; Line 494: Remove "only"; Line 496: Change "is a large of" to "are large"; Line 500-501: Change "SOA often shared higher factions in" to "SOAs often composed higher fractions of"; Line 502: Change "are much less" to "not well"; Line 504: Change "inside" to "on"; Line 505: Change "during" to "on"; Line 505, 506: Remove "and, in fact"; Line 507: Change "could take" to "took"; Line 509: Change "with" to "from"; Line 510: Change "could also compare" to "also compared"; Line 511: Add "use"; Line 512: Change "investigate" to "investigated"; Line 513: Change "the" to "a"; Line 514-515: Change "period II (3-12 November)" to "period (period II; 3-12 November), the"; Line 515: Change "when compared to that" to " compared to those"; Line 516: Change "And their" to "In addition, the"; Line 516: Change "potentials" to "potential"; Line 518: Change "drop" to "decrease"; Line 519: Change "of" to "over"; Line

520: Change "apportioning" to "apportionment"; Line 523: Chang "about" to "of"; Line 524: Change "With" to "Through"; Line 525: Change "of wind directions" to "in wind direction"; Line 526: Change "Total" to "The total"; Line 527: Change "that" to "those"; Line 531: Change "were" to "was"; Line 533: Change "by" to "of"; Line 534: Change "in average due to drops in the percentages by" to "on average due to decreases in the percentages of"; Line 535-536: Change "that accounted" to "accounting"; Line 537: Change "south" to "southern"; Line 538: Change "north" to "northern".

---

## Author Comment (AC1) · 16 Jul 2018

**Author' Response to Referees' Comments**

**Anonymous Referee #1**

Yang et al. present VOC observational results at a rural site in the Beijing metropolitan area before, during, and after a strict emission control implementation for the APEC summit event. They mainly discussed on changes in the VOC composition during three different periods. Then, they move ahead to discuss the potential sources from the PFM analysis and the impacts of VOCs in the context of ozone and aerosol formation rates. Overall the manuscript is very clearly written, and the goals seem to be well achieved.

[1] However, I have a couple of concerns regarding whether this manuscript is well fit in this particular journal. Although the significance in understanding of Chinese pollution problems cannot be overstated considering the large population in China, this manuscript is too narrowly focused on Chinese local air pollution problems without discussion on the bigger context such as regional or global impacts.

**Reply**: Thanks for the insightful comments. Maybe we have not interpreted our findings properly and fully in the manuscript. We think the findings from our study are implicative at least in two aspects:

1) While exposure to indoor air pollution from burning solid fuels (biomass and coal) affects nearly half of the world's population, and household air pollution has been considered as a major environmental cause of death (Martin et al., 2011; Lim et al., 2012; Subramanian, 2014), a previous study (Liu et al., 2016) revealed that in the Beijing-Tianjian-Hebei (BTH) region, residential use of solid fuels might be a major and underappreciated ambient pollution source for $PM_{2.5}$ (particularly BC and OC) during winter heating period based on the Multiresolution Emission Inventory of China (MEIC; www.meicmodel.org) for January and February 2010. Here we demonstrated that, based on our field monitoring, burning solid fuels (mainly coal) to heat homes would also be a major source of volatile organic compounds (VOCs) during winter in the region. As VOCs are important precursors of secondary aerosols and ozone, our study also suggests that cleaner residential energy use for cooking and heating not only benefit lowering indoor air pollution with great health benefits for the residents, but also benefit lowering the exposure to ambient air pollution for a wider range of people.

2) Due to high secondary aerosol contribution to particulate pollution during haze events (Huang et al., 2014), reducing emissions of VOCs as precursors of secondary organic aerosols is important for combating fine particle air pollution and heavy hazes. Enhacing the controls over emissions from vechicles and industry sector would be effective for reducing ambient VOCs, as demonstrated in our study during the APEC before the start of winter heating period. During the winter heating period, since residential coal/biomass burning was found to be a major source for ambient VOCs even in the Beijing metropolitan area, solely enhancing the emisison control in the traffic and industry sectors would be not so effective as did in the non-heating period. This is a important message for regions, particularly less developed regions, to control emisisons of VOCs to combat air pollution due to ozone and $PM_{2.5}$.

Therefore, although we conducted our study at a rural site in Beijing, the findings are not just locally significant but also have important implications for other regions. We have incorporated these aspects into our revised manuscript. We have rewrite our introduction, and modified our conclusions and abstract as well.

[2] Moreover, the scientifically relevant analyses such as ozone forming potential and secondary aerosol forming potential are just adapted from previous publications without detailed discussion whether the method is relevant to this particular photochemical environment.

**Reply**: Thanks. This comment is scientifically very important. It reminds us of remembering that the potentials are related to particular photochemical environments. As for the ozone formation potentials, the Maximum Incremental Reactivity (MIR) scale, originally developed by Carter (1994), has been widely used as a simplified approach to evaluate the relative ground-level ozone impacts of volatile organic compounds. The MIR scale in its nature represents conditions where ambient ozone is most sensitive to changes in VOC emissions, therefore the potentials based on the MIR scale are maximums that can hardly achieved under real atmospheric conditions, particularly depending on the relative availability of NOx (Dondge, 1984; Carter and Atkinson, 1989). However, for the convenience of regulating VOCs based on calculations of their relative ground-level ozone impacts, the metrics used for calculating OFP in the present study have been used worldwide, and therefore we remain the calculation in its present state but indicate in the revised manuscript that it is only a simplified approach.

Organic aerosol formation potentials are comparatively much more complicated. They are largely affected by factors such as the reactivity of the parent compound and volatility of the product species (Odum et al., 1997). The reactivity of the parent species can be directly measured by their reaction rate constants with oxidants. The oxidation products, however, are both numerous and difficult to quantify analytically. Therefore, the SOA yield (Y), defined as mass of SOA formed divided by mass of VOCs reacted, has been used as an indirect measure for a specific VOC species to indicate its ability to form SOA (Odum et al., 1997). This way the secondary organic aerosol formation potentials (SOAFPs) by a mixture of VOCs can be estimated as $\Sigma_i X_i \times Y_i$, where $X_i$ is the mass concentration ($\mu g\ m^{-3}$) and $Y_i$ (%) is the SOA yield of precursor i. SOA yield data have been obtained in controlled smog chamber studies. In this study, the SOA yields are taken from studies by Ng et al (2007), Lim and Ziemann (2009) and Loza et al (2014). As SOA formation depends on nitrogen oxides (NOx) (Ng et al., 2007), SOAFPs are typically calculated under low-NOx and high-NOx conditions, approximating the higher and lower limits, respectively. Although widely used in a lot of literatures, this kind of calculation is also a simplified approach to indicate SOA potentially formed if the observed VOCs are completely oxidized in the atmosphere.

In the present study, we put our focus mainly on how the control measures or human activities would impact the VOCs occurring in the ambient air, so we just followed the widely adopted approaches to indicate their ozone and SOA formation potentials, although they are simplified and even scientifically not solid enough.

[3] I recommend expanding discussion at least to compare VOC speciation from other locations in the region and other metropolitan areas in the world.

**Reply**: Thanks for the suggestion. In the revised manuscript, we have added the comparison as beolw:

"Table S1 shows a comparison of VOCs from our study with those observed at other metropolitan areas in the world. Mixing ratios of VOCs from this study at a rural site in Beijing during period I (23.41 ppb) and period III (21.71 ppbv) were comparable to that in urban Shanghai from January 2007 to March 2010 (Cai et al., 2010), but lower than those in Beijing during June 2008 (Wang et al., 2010), Guangzhou from June 2011 to May 2012 (Zou et al., 2015), Lille, French from May 1997 to April 1999 (Borbon et al., 2002) and Houston in August-September 2006 (Leuchner and

Rappengluck, 2010). Average mixing ratios of VOCs during period II (11.25 ppbv) with enhanced emission control in the present study were significantly lower than those reported in other metropolitan areas. As for the most abundant VOC species including ethane, propane, ethylene, benzene, toluene and ethyne, the mixing ratios of ethane and ethylene at UCAS were similar to that at Beijing during June 2008 (Wang et al., 2010) and urban Guangzhou from June 2011 to May 2012 (Zou et al., 2015), but significantly lower than that in urban Beijing during 2014 APEC (Li et al., 2015). Propane in present study are comparable with that in Hong Kong from September 2002 to August 2003 (Guo et al., 2007) and Lille, French from May 1997 to April 1999 (Borbon et al., 2002), but factors of 2-3 lower than that reported in urban Shanghai from January 2007 to March 2010 (Cai et al., 2010) and Guangzhou from June 2011 to May 2012 (Zou et al., 2015). Mixing ratios of benzene and toluene in Lille, French from May 1997 to April 1999 (Borbon et al., 2002) were over 2 times higher than that in present study. Mixing ratios of ethylene, benzene and toluene in present study were comparable to those observed in Houston during August-September 2006 (Leuchner and Rappengluck, 2010), while ethyne, a tracer of incomplete combustion, had mixing ratios 3-4 times higher than that in Houston."

[Figure]

Figure S1. Comparison of (a) ethane, (b) propane, (c) ethylene, (d) benzene, (e) toluene and (f) ethyne observed at UCAS with those from other studies at metropolitan areas in the world.

[4] In addition, I would recommend more thorough descriptions on ozone formation potential and SOA formation potential particularly in the context of whether the metrics are relevant in this photochemical environment.

**Reply**: As responding to comment [2] above, we fully agree that we should consider if the metrics are relevant in a specific photochemical environment. However, in the present study since we put our focus mainly on how the control measures or human activities would impact the VOCs occurring in the ambient air, so we just followed the widely adopted approaches to indicate their ozone and SOA formation potentials, although they are over simplified and even scientifically not solid enough.

[revised manuscript text omitted]

---

## Author Comment (AC2) · 16 Jul 2018

**Author' Response to Referees' Comments**

**Anonymous Referee #2**

Household air pollution from burning biomass and coal for cooking foods and heating rooms has long been a major environmental problem. Previous studies suggested that the uncontrolled and inefficient combustion of solid fuels for heating in winter also contributed substantially to outdoor $PM_{2.5}$, BC, OC, $SO_2$ and NOx in many regions, such as in the North China Plain (Liu et al., 2016, PNAS). The field observations at the rural site of Beijing in this study further demonstrated that the combustion of solid fuels for heating in winter made remarkable contribution to ambient volatile organic compounds (VOCs). The authors also took advantage of the temporary intervention measures for emission control during the APEC to evaluate the actual effect of the control measures on the ambient VOCs levels through reduction of the source contributions. The comparison between the heating and non-heating periods offered robust results indicating the influential emission from winter heating. Overall, the data quality of this manuscript is quite good, and the interpretation of the results is appropriate and convincing. Therefore, I strongly recommend publication of this manuscript.

**Reply**: Thanks for the comments. We have revised our manuscript with your constructive comments and suggestions as below.

Majors:

[1] As stated above, an important finding of this study is that residential coal burning, especially during winter heating period, could be a major contributor to the ambient VOCs. Apart from the emission factors available for residential coal burning, is it possible for the authors to have the amounts of coals consumed in the residential sector especially during winter and give an in-depth explanation why this source could contribute substantially?

**Reply**: Thanks for the suggestions. The information about coal consumptions and an in-depth explanation to the substantial contribution from residential coal burning have been added into the revised manuscript (section 3.3.2, line 443-459):

"Coal is consumed in residential, industrial and power sectors in Beijing. As showed in Fig. S5a, while annual total coal consumptions dropped rapidly during 2006-2015, the annual residential coal

consumptions remained almost unchanged with their percentages in total coal consumptions rising from 8.7% in 2006 to 23.4% in 2015 (Beijing Municipal Bureau of Statistics, 2016; Yu et al., 2018). As a matter of fact, over 60% of the residential coal consumption occurred in rural areas of Beijing (Fig. S5b), and residential coal is mainly burned in the cool winter season for house heating (Xue et al., 2016). While emission factors of VOCs from residential coal burning have been found to be a factor of 20 greater than those from coal-fired power plants (Liu et al., 2017), the differences in coal quality between the urban and rural areas augment emissions in rural areas: coal used in urban area was entirely anthracite with comparatively much lower emissions of volatiles than other types of coal (Xu et al., 2017); instead only 5-15% of coal used in rural area was anthracite (Xue et al., 2016). Consequently, residential coal combustion could have been a major contributor to the ambient VOCs in rural areas of Beijing during winter."

The coal consumptions in Beijing from 2006 to 2015 were also added to the supplement information as shown below (Fig. S5).

[Figure]

Figure S5. (a) The total coal consumption in residential, industrial and power generation sectors and the percentage of residential coal consumption in total coal consumption in Beijing during 2006-2015; (b) Residential coal consumption in urban and rural areas of Beijing during 2006-2015.

[2] The study suggests clean energy use in residential sector is vital for reducing VOCs in the heavily polluted winter. This aspect should be incorporated into the discussion or the conclusion part of the manuscript.

**Reply**: Thanks. In the revised manuscript we have re-write the introduction part to stress that residential burning of solid fuels for cooking and heating is not only a problem of indoor air

pollution, but also an important source of outdoor air pollutants. In the conclusion part, based on the results that residential burning of solid fuels contributed nearly halves of VOCs in ambient air during wintertime heating, we added:

"However, as observed in this study, even in megacities like Beijing, burning raw coal or biomass for household heating in winter could contribute near half of VOCs in ambient air. If the emission control over residential burning of solid fuels is underappreciated, the intervention measures targeted on traffic and industry sectors would be not so effective in the wintertime heating period as did in non-heating periods either to lower $PM_{2.5}$ as indicated by Liu et al. (2016) or to lower VOCs in ambient air as indicted by this study. If fact, a study by Yu et al. (2018) during the same field campaign of this study demonstrated that, without emission control over residential burning of solid fuels, ambient $PM_{2.5}$-bound toxic polycyclic aromatic hydrocarbons in rural Beijing during the 2014 APEC summit remained unchanged despite of the temporary intervention control measures, and they were largely aggravated after the start of wintertime heating. Therefore, cleaner energy use instead of poor-technology burning of solid fuels household heating would have tremendous health benefits in lowering both indoor and outdoor air pollution particularly in heavily polluted winter. It worth noting that this study was conducted in a rural area of the megacity Beijing. Emission from residential burning of solid fuels would be a source of greater importance and thus deserves more concern in less developed regions."

Minors:

Although I am not a native English speaker, I would say that there is still room for improving English writing of the manuscript. The authors better find a native English speaker to check the English.

**Reply**: Thanks for your careful check and your great patience in listing the errors/mistakes. For the revised manuscript, we have also asked a native English speaker to re-check the English writing.

Line 33-34: Change "during wintertime severe haze events" to "during severe wintertime haze events";

**Reply**: Revised as suggested.

Line 35: Change "comparatively much less" to "not well";

**Reply**: Revised as suggested.

Line 37: Change "inside" to "on";

**Reply**: Revised as suggested.

Line 37: add "the" before "University";

**Reply**: Revised as suggested.

Line 38: Change "northeast" to "northeastern";

**Reply**: Revised as suggested.

Line 39: Remove "that";

**Reply**: Revised as suggested.

Line 39: Change "during" to "on";

**Reply**: Revised as suggested.

Line 40: Add "the" before "air quality";

**Reply**: Revised as suggested.

Line 41: Change "in" to "on";

**Reply**: Revised as suggested.

Line 41: Remove "that";

**Reply**: Revised as suggested.

Line 41: Change "since" to "on";

**Reply**: Revised as suggested.

Line 42: Change "it is" to "this sample collection period provided";

**Reply**: Revised as suggested.

Line 43: Remove "the" before "temporary" and before "wintertime";

**Reply**: Revised as suggested.

Line 44, 45: Add "the" before "temporary";

**Reply**: Revised as suggested.

Line 46: Change "about" to "approximately";

**Reply**: Revised as suggested.

Line 47: Change "that" to "the values";

**Reply**: Revised as suggested.

Line 46-48: Change "that of 23.41 ppb before the APEC (25 October-2 November; Period I) or

21.71 ppb after the APEC (13 November-31 December; Period III)" to "the values of 23.41 ppb in Period I (25 October-2 November) before the APEC and 21.71 ppb in period III (13 November-31 December) after the APEC";

**Reply**: Revised as suggested.

Line 48: Change "Their" to "The";

**Reply**: Revised as suggested.

Line 49: Change "drop" to "decrease";

**Reply**: Revised as suggested.

Line 50: Change "of" to "over";

**Reply**: Revised as suggested.

Line 51: Remove "the" before "southerly";

**Reply**: Revised as suggested.

Line 52: Change "that in the northerly ones during period I, II and III" to "those in northerly air masses during periods I, II and III";

**Reply**: Revised as suggested.

Line 53: Remove "and";

**Reply**: Revised as suggested.

Line 54: Change "south" to "southern";

**Reply**: Revised as suggested.

Line 56: Change "changed" to "the altered";

**Reply**: Revised as suggested.

Line 57, 58: Remove "the" before "Period";

**Reply**: Revised as suggested.

Line 58: Change "emission" to "emissions";

**Reply**: Revised as suggested.

Line 59: Add "the" before "ambient";

**Reply**: Revised as suggested.

Line 60: Change "emission" to "emissions";

**Reply**: Revised as suggested.

Line 60: Remove "they";

**Reply**: Revised as suggested.

Line 61-62: Change "VOCs during the period I, II and III" to "the VOCs during periods I, II and III";

**Reply**: Revised as suggested.

Line 62-63: Change "became the dominant source which accounted for 45.1% of the VOCs" to "became the largest source, accounting for 45.1% of the VOCs";

**Reply**: Revised as suggested.

Line 64-65: Change "with a remarkably lower average contribution percentage (38.2%) in the southerly air masses than that of 48.8% in the northerly air masses" to "with a specifically lower average contribution percentage in southerly air masses (38.2%) than in northerly air masses (48.8%)";

**Reply**: Revised as suggested.

Line 72: Change "emission" to "emissions";

**Reply**: Revised as suggested.

Line 73: Change "benefit improving" to "improve the";

**Reply**: Revised as suggested.

Line 74: Change "an extensive concern" to "a widespread concern";

**Reply**: Revised as suggested.

Line 75, 79: Change "summertime" to "summer";

**Reply**: Revised as suggested.

Line 75: Change "wintertime" to "winter";

**Reply**: Revised as suggested.

Line 80: Remove "as well";

**Reply**: Revised as suggested.

Line 82: Change "comparatively the role of VOCs in the wintertime with $PM_{2.5}$" to "comparatively, the effect of VOCs on wintertime $PM_{2.5}$";

**Reply**: Revised as suggested.

Line 88: Change "in the control of air pollution by $PM_{2.5}$ in wintertime" to "in the control of $PM_{2.5}$ air pollution in winter";

**Reply**: Revised as suggested.

Line 90: Change "under" to "on";

**Reply**: Revised as suggested.

Line 92: Change "vehicle exhausts are" to "vehicle exhaust is an";

**Reply**: Revised as suggested.

Line 96: Change "particularly in north China in wintertime" to "particularly in northern China in winter";

**Reply**: Revised as suggested.

Line 99: Add "the" before "VOCs";

**Reply**: Revised as suggested.

Line 100: Change "is widely occurring in the rural areas" to "widely occur in rural areas";

**Reply**: Revised as suggested.

Line 101-102: Change "how the enhanced emission" to "how enhanced emissions";

**Reply**: Revised as suggested.

Line 109: Change "north" to "northern";

**Reply**: Revised as suggested.

Line 110: Add "the" before "air quality";

**Reply**: Revised as suggested.

Line 111: Remove "so";

**Reply**: Revised as suggested.

Line 111: Remove "when";

**Reply**: Revised as suggested.

Line 112: Change "especially in wintertime with" to "especially in winter, with";

**Reply**: Revised as suggested.

Line 116: Change "kind" to "type";

**Reply**: Revised as suggested.

Line 118: Change "As for" to "For";

**Reply**: Revised as suggested.

Line 121: Change "drops of" to "decreases in";

**Reply**: Revised as suggested.

Line 123: Change "inside" to "on";

**Reply**: Revised as suggested.

Line 124: Add "the" before "total";

**Reply**: Revised as suggested.

Line 124: Add "were" before "reduced";

**Reply**: Revised as suggested.

Line 125: Add "those in" before "the period";

**Reply**: Revised as suggested.

Line 126: Change "about" to "of";

**Reply**: Revised as suggested.

Line 127, 134: Remove "the" before "urban";

**Reply**: Revised as suggested.

Line 127: Remove "entirely";

**Reply**: Revised as suggested.

Line 128: Add "on" before "a regional scale";

**Reply**: Revised as suggested.

Line 132-133: Change "The objectives of present study are" to "The objectives of the present study are as follows";

**Reply**: Revised as suggested.

Line 135, 136: change "crucial" to "the major";

**Reply**: Revised as suggested.

Line 136: Change "wintertime" to "winter";

**Reply**: Revised as suggested.

Line 137: Add "the" before "APEC";

**Reply**: Revised as suggested.

Line 141: Change "_" to "_";

**Reply**: Revised as suggested.

Line 142: Change "inside the campus of" to "on the campus of the";

**Reply**: Revised as suggested.

Line 142: Add "the" before "Huairou";

**Reply**: Revised as suggested.

Line 143-144: Change "The UCAS is located about 60 km northeast of central Beijing and about 150 km northwest of the Tianjin city" to "UCAS is located approximately 60 km northeast of the center of Beijing and approximately 150 km northwest of the city of Tianjin";

**Reply**: Revised as suggested.

Line 145-147: Change "16 meters above ground on the top of a four-story building, about 100 m west of a national road and only 1.5 km far away from the APEC main conference hall" to "16 meters above the ground on the top of a four-story building, approximately 100 m west of a national road and only 1.5 km away from the main APEC conference hall";

**Reply**: Revised as suggested.

Line 152: Remove "of";

**Reply**: Revised as suggested.

Line 153: Add "and" before "one";

**Reply**: Revised as suggested.

Line 154: Add "was" before "less than";

**Reply**: Revised as suggested.

Line 154: Add "a" before "relative";

**Reply**: Revised as suggested.

Line 154: Add "of" before "less";

**Reply**: Revised as suggested.

Line 158: Remove "the time span";

**Reply**: Revised as suggested.

Line 163-164: Change "and average temperature was 11.4, 7.0, and 0.6°C during periods I, II and III" to "and the average temperature was 11.4, 7.0 and 0.6°C during periods I, II and III";

**Reply**: Revised as suggested.

Line 224: Remove "the" before "period";

**Reply**: Revised as suggested.

Line 226: Change "Total" to "The total";

**Reply**: Revised as suggested.

Line 226: Change "inside" to "at";

**Reply**: Revised as suggested.

Line 227: Change "in" to "on";

**Reply**: Revised as suggested.

Line 229: Change "halves of 57.45, 36.17, and 56.56 ppb" to "half the values (57.45, 36.17, and 56.56 ppb)";

**Reply**: Revised as suggested.

Line 231: Remove "both";

**Reply**: Revised as suggested.

Line 234: Add "a" before "more than";

**Reply**: Revised as suggested.

Line 234: Remove "the";

**Reply**: Revised as suggested.

Line 234: Change "about" to "an approximately";

**Reply**: Revised as suggested.

Line 237: Change "or" to "and the";

**Reply**: Revised as suggested.

Line 238: Change "densities" to "density";

**Reply**: Revised as suggested.

Line 240: Change "shared by" to "of";

**Reply**: Revised as suggested.

Line 240: Remove "quite";

**Reply**: Revised as suggested.

Line 243-244: Change "percentages shared by aromatics became lower during period II (12%) when compared to that in period I (21%) or period III (15%)" to "the percentage of aromatics was lower during period II (12%) than during period I (21%) and period III (15%)";

**Reply**: Revised as suggested.

Line 246-247: Change "decreased by 49.0, 32.5, 72.8, and 48.1%, respectively, when compared to those during period I" to "were 49.0, 32.5, 72.8 and 48.1% lower than those during period I, respectively";

**Reply**: Revised as suggested.

Line 247-248: Change "Aromatics evidently had a more substantial drop" to "Aromatics evidently

underwent a larger decrease";

**Reply**: Revised as suggested.

Line 249: Remove "as";

**Reply**: Revised as suggested.

Line 252: Change "ozone formation potentials (OFPs)" to "ozone formation potential (OFP)";

**Reply**: Revised as suggested.

Line 253: Change "in average during periods I, II and III were" to "on average during periods I, II and III was";

**Reply**: Revised as suggested.

Line 255: Change "Their" to "The";

**Reply**: Revised as suggested.

Line 255: Change "potentials (SOAFPs)" to "potential (SOAFP)";

**Reply**: Revised as suggested.

Line 257, 259: Add "the" to "total";

**Reply**: Revised as suggested.

Line 261: Change "VOCs" to "VOC";

**Reply**: Revised as suggested.

Line 262: Change "drop" to "decreases";

**Reply**: Revised as suggested.

Line 264: Change "changed contribution by aromatic" to "the altered contribution of aromatics";

**Reply**: Revised as suggested.

Line 266: Change "condition" to "conditions and from";

**Reply**: Revised as suggested.

Line 267: Change "condition" to "conditions";

**Reply**: Revised as suggested.

Line 270: Change "showed" to "shown";

**Reply**: Revised as suggested.

Line 271: Change "with the increase in" to "with an increase in the";

**Reply**: Revised as suggested.

Line 272: Change "like that" to "as those";

**Reply**: Revised as suggested.

Line 276: Add "the" before "wind";

**Reply**: Revised as suggested.

Line 280: Change "showed" to "shows";

**Reply**: Revised as suggested.

Line 281: Change "the" to "a";

**Reply**: Revised as suggested.

Lines 282-285: "It clearly demonstrated that the mixing ratios of VOCs increased rapidly, and the back trajectories indicated that air masses changed from northerly to southerly and then declined sharply while the air masses changed back from southerly to northerly", rewrite the sentences.

**Reply**: Revised as suggested.

Lines 285-286: Change "The southern areas of UCAS are the central Beijing with stronger emissions" to "The southern areas of UCAS are in central Beijing where emissions are stronger";

Line 286: Add "that";

**Reply**: Revised as suggested.

Line 287: Change "increase of" to "increase in the";

**Reply**: Revised as suggested.

Line 288: Change "of source regions" to "in the source region";

**Reply**: Revised as suggested.

Line 289: Change "showed" to "shown";

**Reply**: Revised as suggested.

Line 291: Change "changed" to "altered";

**Reply**: Revised as suggested.

Line 295: Add "and" before "2)";

**Reply**: Revised as suggested.

Line 296: should be ": : :Mongolia and quickly: : :";

**Reply**: Revised as suggested.

Line 297: Change "emission" to "emissions";

**Reply**: Revised as suggested.

Line 302: Change "period" to "periods";

**Reply**: Revised as suggested.

Line 303: Change "that" to "those";

**Reply**: Revised as suggested.

Line 304: Change "OFPs in the southerly air masses were" to "the OFP in southerly air masses was";

**Reply**: Revised as suggested.

Line 305: Change "and SOAFPs in the" to "and the SOAFP in";

**Reply**: Revised as suggested.

Line 305: Change "were" to "was";

**Reply**: Revised as suggested.

Line 308: Change "This indicates that the north and south" to "These results indicate that the northern and southern";

**Reply**: Revised as suggested.

Line 313-314: Change "OFPs and SOAPFs" to "OFP and SOAFP";

**Reply**: Revised as suggested.

Line 314: Add "that" before "the changes";

**Reply**: Revised as suggested.

Line 315: Remove "to";

**Reply**: Revised as suggested.

Line 316: Change "when compared to that during period I average" to "compared to that during period I, the average";

**Reply**: Revised as suggested.

Line 319: Change "OFPs decreased by 48.1% and SOAFPs" to "the OFP decreased by 48.1% and the SOAFP";

**Reply**: Revised as suggested.

Line 320: Change "when compared to that" to "compared to those";

**Reply**: Revised as suggested.

Line 321: Add "the" before "average";

**Reply**: Revised as suggested.

Line 323: Remove "when";

**Reply**: Revised as suggested.

Line 323-324: Change "OFPs decreased by 48.9% and SOAFPs decreased by over 70% during period II relative to period I" to "the OFP decreased by 48.9%, and the SOAFP decreased by over 70% during period II relative to those in period I";

**Reply**: Revised as suggested.

Line 325: Add "a" before "more";

**Reply**: Revised as suggested.

Line 326: Change "emission control in" to "control over emissions from";

**Reply**: Revised as suggested.

Line 327: Change "less changes in mixing ratios" to "decreased changes in the mixing ratios";

**Reply**: Revised as suggested.

Line 328: Change "of" to "over";

**Reply**: Revised as suggested.

Line 331: Change "that" to "those";

**Reply**: Revised as suggested.

Line 331: Change "This difference in the increase rates" to "These different increases";

**Reply**: Revised as suggested.

Line 332: Change "are" to "were";

**Reply**: Revised as suggested.

Line 333: Change "heating supply was only available since" to "heat sources were only available after";

**Reply**: Revised as suggested.

Line 334: Change "already" to "have";

**Reply**: Revised as suggested.

Line 336: Change "apportioning" to "apportionment";

**Reply**: Revised as suggested.

Line 338: Add "the" before "campaign";

**Reply**: Revised as suggested.

Lines 338-339: Change "might be resulted from changed contribution by emission sources, such like" to "might have resulted from the altered contributions from emission sources, such as";

**Reply**: Revised as suggested.

Line 341: add "the" before "characteristic";

**Reply**: Revised as suggested.

Line 344: Change "in average during period" to "on average during periods";

**Reply**: Revised as suggested.

Line 345: Change "were approaching" to "approached";

**Reply**: Revised as suggested.

Line 347, 351: Remove "the" before "period";

**Reply**: Revised as suggested.

Line 348: Remove ", which is characteristic of vehicular exhaust";

**Reply**: Revised as suggested.

Line 349: Add "the" before "incomplete";

**Reply**: Revised as suggested.

Line 354: Change "in" to "on";

**Reply**: Revised as suggested.

Line 355: Change "that coal burning contributed more" to "an increased contribution of coal burning";

**Reply**: Revised as suggested.

Line 357: Change "emission" to "emissions";

**Reply**: Revised as suggested.

Line 359: Change "are mainly from" to "mainly originate from";

**Reply**: Revised as suggested.

Line 361: Change "were" to "are";

**Reply**: Revised as suggested.

Line 364: Change "when compared to that of" to "than the values of";

**Reply**: Revised as suggested.

Line 365: Change "or" to "and of";

**Reply**: Revised as suggested.

Line 366: Change "This drop in aromatics/CO ratios" to "This decrease in the ratios of aromatics to CO";

**Reply**: Revised as suggested.

Line 367: Change "of" to "over";

**Reply**: Revised as suggested.

Line 370, 371: Add "those in" before "period I";

**Reply**: Revised as suggested.

Line 371-372: Change "Apparently larger decrease in TEX/CO ratios in the northerly air masses reflected" to "A larger decrease in the TEX/CO ratios in northerly air masses reflects the fact that";

**Reply**: Revised as suggested.

Line 374: Change "3.3.2 Source Apportioning by PMF" to "3.3.2 Source Apportionment by PMF";

**Reply**: Revised as suggested.

Line 375: Change "Thirty-five" to "The 35";

**Reply**: Revised as suggested.

Line 375: Add "and" before "ethyne";

**Reply**: Revised as suggested.

Line 376: Change "sources" to "source";

**Reply**: Revised as suggested.

Line 377: Add "use with" before "the PMF";

**Reply**: Revised as suggested.

Line 385: Change "the gasoline vehicle emission" to "gasoline vehicle emissions";

**Reply**: Revised as suggested.

Line 386: should be "trichloroethylene and tetrachloroethylene";

**Reply**: Revised as suggested.

Line 387: Change "of" to "by";

**Reply**: Revised as suggested.

Line 388: Change "manufacturing industrials" to "industrial manufacturing";

**Reply**: Revised as suggested.

Line 389-390: Change "by industries for make" to "in industry to prepare";

**Reply**: Revised as suggested.

Line 390: Add "during the" before "production";

**Reply**: Revised as suggested.

Line 392: Change "emission" to "emissions";

**Reply**: Revised as suggested.

Line 393: Change "a larger percentage of the" to "larger percentages of";

**Reply**: Revised as suggested.

Line 394: Chang "It is known that TEX are" to "TEX is known to be";

**Reply**: Revised as suggested.

Line 396: Change "They are also" to "These compounds are also the";

**Reply**: Revised as suggested.

Line 396-397: Change "auto factory painting and building coating" to "automobile factories, paint and building coatings";

**Reply**: Revised as suggested.

Line 398: Change "as" to "to be";

**Reply**: Revised as suggested.

Line 403-404: Change "top 3 species of" to "the top 3 species emitted during";

**Reply**: Revised as suggested.

Line 405: Change "were" to "are";

**Reply**: Revised as suggested.

Line 405-406: Change "and aromatics like benzene" to "as well as aromatics such as benzene";

**Reply**: Revised as suggested.

Line 406-407: Change "So factor 5 is related to the coal/biomass burning" to "Thus, factor 5 is related to coal/biomass burning";

**Reply**: Revised as suggested.

Line 408: Change "period" to "periods";

**Reply**: Revised as suggested.

Line 409: Add "the" before "VOCs";

**Reply**: Revised as suggested.

Line 411: Add the "the" before "temporary" and remove "the" before "period";

**Reply**: Revised as suggested.

Line 412: Change "by" to "of";

**Reply**: Revised as suggested.

Line 413: Change "drop" to "decrease";

**Reply**: Revised as suggested.

Line 414: Change "by" to "from";

**Reply**: Revised as suggested.

Line 415: Change "Quite similar" to "Similar";

**Reply**: Revised as suggested.

Line 417: Remove "the";

**Reply**: Revised as suggested.

Line 418: Change "from" to "on";

**Reply**: Revised as suggested.

Line 419: Change "emission" to "emissions";

**Reply**: Revised as suggested.

Line 420: Add "the" before "VOCs";

**Reply**: Revised as suggested.

Line 421: Change "were showed" to "are shown";

**Reply**: Revised as suggested.

Line 422: Change "by" to "of";

**Reply**: Revised as suggested.

Line 422: Change "with the" to "with the increase in";

**Reply**: Revised as suggested.

Line 423: Remove "instead";

**Reply**: Revised as suggested.

Line 427: Change "consumptions" to "consumption";

**Reply**: Revised as suggested.

Line 429: Change "The residential coal combustion is prevailing" to "Residential coal combustion is primarily carried out";

**Reply**: Revised as suggested.

Line 429: Change "by using" to "with";

**Reply**: Revised as suggested.

Line 430: Add "the" before "rural";

**Reply**: Revised as suggested.

Line 431: Change "wintertime" to "winter";

**Reply**: Revised as suggested.

Line 431: Remove "the";

**Reply**: Revised as suggested.

Line 431: Change "accounts" to "accounted";

**Reply**: Revised as suggested.

Line 433: Change "contribute predominately to" to "have been the predominant contributor to the";

**Reply**: Revised as suggested.

Line 437, 438, 444, 446, 449: Remove "the" before "period";

**Reply**: Revised as suggested.

Line 437, 440, 441: Change "by" to "from";

**Reply**: Revised as suggested.

Line 441: Change "reduced" to "decreased";

**Reply**: Revised as suggested.

Line 443: Change "showed" to "made";

**Reply**: Revised as suggested.

Line 443: Change "period" to "periods";

Line 447: Change "account" to "accounted";

**Reply**: Revised as suggested.

Line 451: Change "period" to "periods";

**Reply**: Revised as suggested.

Line 451: Remove "respectively";

**Reply**: Revised as suggested.

Line 452: Change "emission (gasoline and diesel vehicles) was" to "emissions (gasoline + diesel) were";

**Reply**: Revised as suggested.

Line 453: Change "VOCs during the period" to "the VOCs during periods";

**Reply**: Revised as suggested.

Line 454: Remove "instead";

**Reply**: Revised as suggested.

Line 456: Change "period" to "periods";

**Reply**: Revised as suggested.

Line 459: Change "Contributions" to "The contributions";

**Reply**: Revised as suggested.

Line 460: Change "were" to "are";

**Reply**: Revised as suggested.

Line 460-461: Change "was mainly coming" to "mainly originated";

**Reply**: Revised as suggested.

Line 467, 468, 470: Change "of" to "in";

**Reply**: Revised as suggested.

Line 472: Change "emission" to "emissions";

**Reply**: Revised as suggested.

Line 474: Change "3.3.3 Source contributions to SOAFPs" to "3.3.3 Source contributions to the SOAFP";

**Reply**: Revised as suggested.

Line 475: Change "apportioning" to "apportionment";

**Reply**: Revised as suggested.

Line 475: Change "SOAFPs by" to "the SOAFP of";

**Reply**: Revised as suggested.

Line 476: Change "showed" to "shown";

**Reply**: Revised as suggested.

Line 477: Change "condition" to "conditions";

**Reply**: Revised as suggested.

Line 477: Change "SOAFPs by solvent use were much higher than that by" to "SOAFP of solvent use was much higher than that of";

**Reply**: Revised as suggested.

Line 478: Change "were" to "was";

**Reply**: Revised as suggested.

Line 479, 481: Change "period" to "periods";

**Reply**: Revised as suggested.

Line 482: Change "of SOAFPs" to "in SOAFP";

**Reply**: Revised as suggested.

Line 483: Change "by" to "from";

**Reply**: Revised as suggested.

Line 485: Change "condition," to "conditions, the";

**Reply**: Revised as suggested.

Line 485: Change "of" to "in";

**Reply**: Revised as suggested.

Line 486: Change "by" to "from";

**Reply**: Revised as suggested.

Line 490: Change "or" to "and";

**Reply**: Revised as suggested.

Line 490: Change "from" to "during";

**Reply**: Revised as suggested.

Line 494: Change "SOAFPs" to "the SOAFP";

**Reply**: Revised as suggested.

Line 494: Remove "only";

**Reply**: Revised as suggested.

Line 496: Change "is a large of" to "are large";

**Reply**: Revised as suggested.

Line 500-501: Change "SOA often shared higher factions in" to "SOAs often composed higher fractions of";

**Reply**: Revised as suggested.

Line 502: Change "are much less" to "not well";

**Reply**: Revised as suggested.

Line 504: Change "inside" to "on";

**Reply**: Revised as suggested.

Line 505: Change "during" to "on";

**Reply**: Revised as suggested.

Line 505, 506: Remove "and, in fact";

**Reply**: Revised as suggested.

Line 507: Change "could take" to "took";

**Reply**: Revised as suggested.

Line 509: Change "with" to "from";

**Reply**: Revised as suggested.

Line 510: Change "could also compare" to "also compared";

**Reply**: Revised as suggested.

Line 511: Add "use";

**Reply**: Revised as suggested.

Line 512: Change "investigate" to "investigated";

**Reply**: Revised as suggested.

Line 513: Change "the" to "a";

**Reply**: Revised as suggested.

Line 514-515: Change "period II (3-12 November)" to "period (period II; 3-12 November), the";

**Reply**: Revised as suggested.

Line 515: Change "when compared to that" to "compared to those";

**Reply**: Revised as suggested.

Line 516: Change "And their" to "In addition, the";

**Reply**: Revised as suggested.

Line 516: Change "potentials" to "potential";

**Reply**: Revised as suggested.

Line 518: Change "drop" to "decrease";

**Reply**: Revised as suggested.

Line 519: Change "of" to "over";

**Reply**: Revised as suggested.

Line 520: Change "apportioning" to "apportionment";

**Reply**: Revised as suggested.

Line 523: Chang "about" to "of";

**Reply**: Revised as suggested.

Line 524: Change "With" to "Through";

**Reply**: Revised as suggested.

Line 525: Change "of wind directions" to "in wind direction";

**Reply**: Revised as suggested.

Line 526: Change "Total" to "The total";

**Reply**: Revised as suggested.

Line 527: Change "that" to "those";

**Reply**: Revised as suggested.

Line 531: Change "were" to "was";

**Reply**: Revised as suggested.

Line 533: Change "by" to "of";

**Reply**: Revised as suggested.

Line 534: Change "in average due to drops in the percentages by" to "on average due to decreases in the percentages of";

**Reply**: Revised as suggested.

Line 535-536: Change "that accounted" to "accounting";

**Reply**: Revised as suggested.

Line 537: Change "south" to "southern";

**Reply**: Revised as suggested.

Line 538: Change "north" to "northern".

**Reply**: Revised as suggested.

References:

Beijing Municipal Bureau of Statistics (BMBS).: Bejing Statistical Yearbook 2016. China Statistics Press, Beijing, 2016.

Liu, C. T., Zhang, C. L., Mu, Y. J., Liu, J. F., and Zhang, Y. Y.: Emission of volatile organic compounds from domestic coal stove with the actual alternation of flaming and smoldering combustion processes, Environ. Pollut., 221, 385-391, http://dx.doi.org/10.1016/j.envpol.2016.11.089, 2017.

Liu, J., Mauzerall, D. L., Chen, Q., Zhang, Q., Song, Y., Peng, W., Klimont, Z., Qiu, X. H., Zhang, S. Q., Hu, M., Lin, W. L., Smith, K. R., and Zhu, T.: Air pollutant emissions from Chinese

households: A major and underappreciated ambient pollution source, P. Natl. Acad. Sci. USA., 113, 7756-7761, http://dx.doi.org/10.1073/pnas.1604537113, 2016.

Xu, J. Y., Zhuo, J. K., Zhu, Y. N., Pan, Y., and Yao, Q.: Analysis of volatile organic pyrolysis products of bituminous and anthracite coals with single-photon ionization time-of-flight mass spectrometry and gas chromatography/mass spectrometry, Energ Fuel, 31, 730-737, https://doi.org/10.1021/acs.energyfuels.6b02335, 2017.

Xue, Y. F., Zhou, Z., Nie, T., Wang, K., Nie, L., Pan, T., Wu, X. Q., Tian, H. Z., Zhong, L. H., Li, J., Liu, H. J., Liu, S. H., and Shao, P. Y.: Trends of multiple air pollutants emissions from residential coal combustion in Beijing and its implication on improving air quality for control measures, Atmospheric Environment, 142, 303-312, 10.1016/j.atmosenv.2016.08.004, 2016.

Yu, Q. Q., Yang, W. Q., Zhu, M., Gao, B., Li, S., Li, G. H., Fang, H., Zhou, H. S., Zhang, H. N., Wu, Z. F., Song, W., Tan, J. H., Zhang, Y. L., Bi, X. H., Chen, L. G., and Wang, X. M.: Ambient $PM_{2.5}$-bound polycyclic aromatic hydrocarbons (PAHs) in rural Beijing: Unabated with enhanced temporary emission control during the 2014 APEC summit and largely aggravated after the start of wintertime heating, Environ. Pollut., 238, 532-542, http://dx.doi.org/10.1016/j.envpol.2018.03.079, 2018.